

# Simultaneous and synergistic profiling of cloud and drizzle properties using ground-based observations

Stephanie P. Rusli[1,2], David P. Donovan[2], and Herman W. J. Russchenberg[1]

[1]Department of Geoscience and Remote Sensing, Faculty of Civil Engineering and Geosciences, TU Delft, Delft, The Netherlands
[2]Royal Netherlands Meteorological Institute (KNMI), De Bilt, The Netherlands

*Correspondence to:* S. P. Rusli (s.rusli-1@tudelft.nl)

**Abstract.** Despite the importance of radar reflectivity ($Z$) measurements in the retrieval of liquid water cloud properties, it remains non-trivial to interpret $Z$ due to the possible presence of drizzle droplets within the clouds. So far, there has been no published work that utilizes $Z$ to identify the presence of drizzle above the cloud base in an optimized and a physically-consistent manner. In this work, we develop a retrieval technique that exploits the synergy of different remote sensing systems to

carry out this task and to subsequently profile the microphysical properties of the cloud and drizzle in a unified framework. This is accomplished by using ground-based measurements of $Z$, lidar attenuated backscatter below as well as above the cloud base, and microwave brightness temperatures. Fast physical forward models coupled to cloud and drizzle structure parametrization are used in an optimal estimation type framework in order to retrieve the best-estimate for the cloud and drizzle property profiles. The cloud retrieval is first evaluated using synthetic signals generated from large-eddy simulation output to verify the

forward models used in the retrieval procedure and the vertical parametrization of the liquid water content. From this exercise it is found that, on average, the cloud properties can be retrieved within 5% of the mean truth. The full cloud-drizzle retrieval method is then applied to a selected ACCEPT campaign dataset collected in Cabauw, The Netherlands. An assessment of the retrieval products is performed using three independent methods from the literature, each was specifically developed to retrieve only the cloud properties, the drizzle properties below the cloud base, or the drizzle fraction within the cloud, respectively. One-

to-one comparisons, taking into account the uncertainties or limitations of each retrieval, show that our results are generally consistent with what is derived using the three independent methods.

## 1   Introduction

Low-level liquid water clouds are known to have a large aerial extent (Hartmann et al., 1992) and consequently a strong impact on the Earth's energy balance (Ramanathan et al., 1989; Slingo, 1990). Observations of these clouds to characterize the mi-

crophysical and radiative processes are therefore needed for climate studies. One important aspect of such observations is the presence of drizzle, which is found to be a common occurence in stratocumulus clouds (Fox and Illingworth, 1997). Drizzle alters the cloud droplet spectra and thus the microphysical structure and radiative properties of the clouds (Feingold et al., 1997; vanZanten et al., 2005). Most notably, drizzle is thought to play a significant role in determining the cloud lifetime (Albrecht, 1989). Additionally, drizzle within the cloud complicates matters by dominating the radar reflectivity signal. Accurately



separating the drizzle contribution from the cloud contribution to the received radar signal is necessary to properly derive the cloud and drizzle properties.

Since liquid water clouds tend to settle at relatively low altitudes in the atmosphere, it is easier to observe them from the surface than from space. Ground based remote sensing systems have the potential to deliver high-resolution time-series data to evaluate and monitor cloud and drizzle properties on a regional scale. A synergystic way in utilizing different remote sensors is a powerful approach that has been widely used to provide a more complete and comprehensive view of these clouds. Active sensors operating in different frequency windows such as radar and lidar provide complementary information on the clouds vertical structure since they 'see' different parts of the cloud (Donovan and van Lammeren, 2001). Microwave radiometers that measure the accumulated radiation along a column provide a particularly accurate way to derive the liquid water path of clouds (Westwater 1978; Peter and Kämpfer 1992).

Various methods that exploit sensor synergy to profile microphysical properties of the liquid water cloud have been developed (Frisch et al. 1995a; Austin and Stephens 2001; McFarlane et al. 2002; Löhnert et al. 2001; Brandau et al. 2010; Martucci and O'Dowd 2011). However, these methods either avoid, ignore or do not fully capture the presence of drizzle. Other techniques that focus on drizzle retrieval are limited in their application to the region below the cloud base (O'Connor et al., 2005; Westbrook et al., 2010). Retrieving the properties of drizzle that is interspersed within the cloud is indeed more difficult. A few hundred meters into the cloud, lidar backscatter signal no longer carries useful information due to the strong attenuation by cloud droplets. While radar has the capability to penetrate deeper into the cloud, the radar reflectivity is known to be sensitive to drizzle droplets that are larger in size as compared to the cloud droplets. Since the observed reflectivity contains contributions from both cloud and drizzle droplets, its interpretation is not straightforward.

Fielding et al. (2015) set a precedence by jointly retrieving cloud and drizzle properties using ground-based radar, lidar and sun photometer observations. This retrieval, however, departs from the assumption that drizzle is present only when the maximum observed reflectivity in a given column exceeds a single threshold value. While the existence of such a reflectivity threshold is supported by many observational studies, the empirically-determined value differs among these studies and can span quite a wide range. Sauvageot and Omar (1987); Frisch et al. (1995b); Mace and Sassen (2000) suggest different $Z$ thresholds in the range of -20 and -15 dBZ. Baedi et al. (2002) showed that the reflectivity due to a non-drizzle component of the cloud reaches a maximum at about -20 dBZ while that of the drizzle component does not get lower than about -10 dBZ, leaving on average a $\sim 10$ dBZ reflectivity gap between the drizzle-contaminated and drizzle-free droplet spectrum. Furthermore, Wang and Geerts (2003) demonstrate that the value of this threshold varies with altitude within the cloud layer and it can increase from around -25 dBZ near the cloud base to about -12 dBZ close to the cloud top. A theoretical approach by Liu et al. (2008) reveals a dependence of the threshold value on the droplet number concentration, a finding that compares favorably with observations. In remote sensing applications, where droplet concentration is one of the unknown variables to retrieve, setting a single $Z$ threshold value in advance may lead to an unaccounted bias in the retrieval.

In this work we develop a retrieval technique that combines ground-based radar, lidar and microwave radiometer (MWR) measurements to simultaneously profile the cloud and drizzle properties without placing apriori constraints on the presence of drizzle droplets within the cloud. There is no predefined reflectivity threshold and so drizzle is always assumed to be



present until that possibility is excluded by the best-fit to the data. The MWR brightness temperature and the lidar attenuated backscatter up to a few hundred meters above the cloud base provide much of the constraint on the cloud component, whereas the radar reflectivity is used to then infer the drizzle contribution. Drizzle droplets are set apart from the cloud droplets through the use of a critical effective radius threshold in the algorithm. This choice of threshold is motivated by the recognition that a

characteristic or critical droplet radius exists, above which intense droplet coalescense triggers rapid drizzle formation. This radius is found to be 12-14 microns as shown by satelite and ground-based observations (Suzuki et al., 2010; Rosenfeld, 2000; Rosenfeld and Gutman, 1994), aircraft measurements (Gerber, 1996; Boers et al., 1998; Freud and Rosenfeld, 2012) and numerical simulations (Magaritz et al., 2009; Pinsky and Khain, 2002; Benmoshe et al., 2012). This retrieval technique allows us to retrieve not only drizzle microphysical properties below the cloud base but also within the cloud at the same time. We

apply this algorithm to synthetic signals for a test case, as well as to observational data collected in the Fall of 2014 as part of the ACCEPT field campaign in Cabauw, The Netherlands. The retrieved cloud and drizzle products from the ACCEPT dataset are evaluated against the results of three independent retrieval methods that use, respectively, the lidar depolarization signal (Donovan et al., 2015), the lidar attenuated backscatter and radar Doppler spectral moments (O'Connor et al., 2005), and the radar Doppler spectra (Kollias et al., 2011a, b; Luke and Kollias, 2013) as their main tools.

The remainder of this paper is organized as follows. Section 2 describes the retrieval procedure in detail, including the theoretical assumptions and the forward models. The test application of this technique to synthetic data based on the large-eddy simulation output is presented in Section 3. In Section 4, we perform the cloud and drizzle retrieval on a ground-based dataset. The retrieval products are then evaluated through comparisons with results from three independent retrieval techniques in Section 5. To conclude the paper, a summary is provided in Section 6.

## 2    Retrieval technique

The target group for this retrieval technique is single-layered liquid water clouds. Heavy precipitation events are avoided and not retrieved. The retrieval products include the optical extinction coefficient, liquid water content, number concentration and the effective radius of both the cloud and the drizzle components separately, as a function of height.

### 2.1    Theoretical basis and parametrization

### 2.1.1    Cloud and drizzle droplet size distribution

We treat the cloud and the drizzle droplets as two separate entities and they are assigned independent and unimodal droplet size distribution (DSD) functions, the combination of which results in a bimodal distribution. Here we assume that the number density of the cloud and drizzle droplets as a function of their size can be described by the a generalized gamma distribution (Walko et al., 1995):

$$n(r) = \frac{N}{r_n \Gamma(\nu)} \left( \frac{r}{r_n} \right)^{\nu-1} \exp(\frac{-r}{r_n}),$$   (1)





where $N = \int_0^\infty n(r)dr$ is the total number concentration, $r_n$ is the droplet characteristic radius and $\nu$ is the shape parameter. The moments of this DSD,

$$\langle r^k \rangle = \frac{\int_0^\infty r^k n(r)dr}{\int_0^\infty n(r)dr},$$

are central in defining and deriving the physical properties of the cloud and the drizzle as listed below.

    – Effective radius $r_e$:

$$r_e = \frac{\langle r^3 \rangle}{\langle r^2 \rangle} = r_n(\nu + 2). \tag{2}$$

    – Extinction coefficient $\alpha$:

$$\alpha = \int_0^\infty Q_{\text{ext},\lambda}(r)\pi r^2 n(r)dr \approx 2\pi N \langle r^2 \rangle, \tag{3}$$

    where the extinction efficiency $Q_{\text{ext},\lambda}(r) \approx 2$ assuming that the droplets are much larger than the wavelength $\lambda$ of the

10    incident light.

    – Liquid water content LWC:

$$\text{LWC} = \frac{4}{3}\pi\rho_w N \langle r^3 \rangle. \tag{4}$$

In addition, the sixth moment of the distribution function delivers the radar reflectivity factor $Z$ by virtue of Rayleigh approximation, which is valid in the case of scattering of particles whose size is small compared to the radar wavelength. The exact

15  expression is:

$$Z = 2^6 \int_0^\infty r^6 n(r)dr = 64N\langle r^6 \rangle. \tag{5}$$

Moreover, the moments of the DSD are assumed to be related to each other such that:

$$\langle r^a \rangle = k_{ab}\langle r^b \rangle^{a/b},$$

in which $k_{ab}$ is a function of shape parameter $\nu$. Using the property of the gamma function we derive, for instance,

$$k_{23}^3 = \frac{\nu(\nu+1)}{(\nu+2)^2}, \tag{6}$$

$$k_{36}^2 = \frac{\nu(\nu+1)(\nu+2)}{(\nu+3)(\nu+4)(\nu+5)}, \tag{7}$$

that allow one to relate LWC to $\alpha$ ($k_{23}$) or $Z$ ($k_{36}$).





### 2.1.2 Cloud structure

To profile the cloud, we adopt an approximation for LWC vertical profile introduced in Boers et al. (2006). Here we repeat what is necessary and adjust some of the notation.

Near the cloud base, LWC is assumed to vary linearly with height (i.e. constant lapse rate). Deeper into the cloud, entrainment

leads to a decrease in the LWC lapse rate. The actual LWC at a given height z (measured from the cloud base) is related to the adiabatic value through a subadiabatic fraction $f(z)$ such that:

$$\text{LWC(z)} = f(z)\text{LWC}_{\text{ad}}(z) = f(z)\rho_a A_{ad} z. \tag{8}$$

$\rho_w$ is the density of water and $\text{LWC}_{\text{ad}}$ the adiabatic LWC. $\rho_a$ and $A_{\text{ad}}$ are the density of air and the adiabatic lapse rate of the liquid water content mixing ratio, respectively; both are a function of the temperature and pressure at the cloud base. The

subadiabatic fraction changes as a function of height and is governed by two variables $W$ and $H$:

$$f(z) = \left[1 - \exp\left(-W\hat{h}\right)\right]\left[1 - \frac{\exp\left(-\hat{h}(1-\hat{z})\right)}{1 - \exp\left(-\hat{h}\right)} + \frac{\exp(-\hat{h})}{1 - \exp\left(-\hat{h}\right)}\right], \tag{9}$$

with $\hat{h} = (z_{\text{ct}} - z_{\text{cb}})/H$ and $\hat{z} = z/(z_{\text{ct}} - z_{\text{cb}})$. The subscript 'ct' and 'cb' denote cloud top and cloud base, $W$ represents the vertical weight of the liquid water distribution and the relaxation length scale $H$ indicates how much the liquid water content departs from adiabaticity. The smaller $W$ is, the more liquid water there is close to the cloud top. The smaller $H$ is, the closer

the actual LWC becomes to the adiabatic profile.

Boers et al. (2006) consider two mixing scenarios to describe the vertical variation in $f(z)$, namely inhomogeneous and homogeneous mixing. In the first one, the variation of $f(z)$ is attributed to the vertical change in $N$:

$$N(z) = f(z)N_{\text{ad}} \tag{10}$$

where $N_{\text{ad}}$ is the adiabatic value of $N$. In the homogeneous mixing case, evaporation causes the droplet sizes to decrease while

preserving the total number of droplets:

$$N(z) = N_{\text{ad}} \tag{11}$$

Once the shape parameter $\nu$, LWC and $N$ are known as a function of height, one can readily derive $r_e$ and $\alpha$ profiles. For a given shape parameter and LWC vertical profile, the two mixing models result in different vertical profiles of the number concentration, effective radius and extinction coefficient. Both mixing scenarios are implemented in the retrieval algorithm,

but for the retrieval in this paper we assume for simplicity that $N$ is constant with height, that is, adopting the homogeneous mixing case.

### 2.1.3 Drizzle structure

The drizzle signature is strongly imprinted in radar reflectivity measurements, making $Z$ indispensable for drizzle retrieval. Owing to the proportionality between the moments of the DSD, the observed reflectivity is related to the drizzle microphysical





properties and the vertical shape of $Z$ can be used to profile drizzle. This is especially true below the cloud base where drizzle is isolated from the cloud and $Z$ is related to drizzle alone.

At the very early stages of drizzle formation, when drizzle is still contained within the cloud and there are no detected drizzle droplets falling from the cloud, $f(z)$ (eq. 9) is used to also describe how drizzle LWC varies with height. Using equations 4, 5 and 7, $r_e$ can be written in terms of $\nu, Z$ and LWC:

$$r_e^3 = \frac{\pi \rho_z Z}{48 \text{LWC}} \frac{(\nu+2)^3}{(\nu+3)(\nu+4)(\nu+5))} \tag{12}$$

from which $N$ and $\alpha$ can be computed.

As drizzle starts to grow and leave the cloud, we use a different drizzle parametrization. The vertical profile of the drizzle effective radius above the cloud base is parametrized via an exponential function:

$$r_e(z) = r_{\text{e,cb}} \exp\left(\frac{k_1 (z - z_{\text{cb}})}{z_{\text{dt}} - z_{\text{cb}}}\right)^{-0.5}. \tag{13}$$

$r_{\text{e,cb}}$ is the value of drizzle effective radius at cloud base, $k_1$ describes the rate at which $r_e$ decreases towards cloud top. The subscript 'dt' denotes drizzle top. The choice of such an exponential function is motivated by the results of in-situ drizzle measurements showing that within the cloud, the drizzle effective radius displays an exponential-like increase towards the cloud base (Wood, 2005a; Lu et al., 2009).

Below the cloud base, drizzle droplets are not expected to keep growing. Instead, they are assumed to evaporate and shrink. In this region, the parametrization of $r_e$ is based on a simple power-law:

$$r_e(z) = r_{\text{e,cb}} \left(\frac{z - z_{\text{db}}}{z_{\text{cb}} - z_{\text{db}}}\right)^{k_2}, \tag{14}$$

with $k_2$ describing the rate at which $r_e$ decreases from the cloud base to the drizzle base (denoted by the subscript 'db').

In the retrieval, the two parameters $k_1$ and $k_2$ are positive and are solved using values of $r_e$ at three different heights: below, at and above the cloud base (see Section 2.3.2). The droplet size information at the cloud base is crucial since it acts as a scaling factor and is the point where these two functions meet. Once the vertical profiles of $\nu, Z$ and $r_e$ are specified, one can derive LWC, $\alpha$ and $N$ as a function of height.

## 2.2 Forward models

Based on the assumptions and parametrizations described in the previous section, one can construct a theoretical cloud and drizzle model. Forward models are then applied to map this theoretical construction to predicted observables that can be compared with measured signals.

### 2.2.1 Radar reflectivity

Equation (5) relates the cloud/drizzle microphysical properties to the radar reflectivity. The equation assumes the validity of Rayleigh approximation. For comparison with the observed reflectivity, the contribution from the cloud and drizzle must be





added and attenuation effects have to be incorporated. The observed reflectivity that we use here is taken the from the Cloudnet categorization product (Illingworth et al., 2007) and has been corrected for two-way attenuation due to atmospheric gases and in some cases, liquid water. Since the liquid water attenuation is dependent on the availability or the reliability of liquid water path measurements, it is not consistently applied to radar pixels containing cloud/drizzle droplets in the Cloudnet algorithm.

For this reason, we recover the measured reflectivity before the liquid, but after the gas, attenuation correction $Z_{\mathrm{obs}}$ using the information provided in the same Cloudnet product. The liquid attenuation is then incorporated in the forward model to compute

$$Z_{\mathrm{fm}} = (Z_{\mathrm{cld}} + Z_{\mathrm{dzl}}) \exp(-2\tau).\tag{15}$$

Hereinafter, the subscripts 'cld' and 'dzl' refer to the cloud and the drizzle components, respectively. The optical depth $\tau$ is

calculated from cloud and drizzle LWC using the approximation for attenuation coefficient given in Liebe et al. (1989). $Z_{\mathrm{fm}}$ is compared to $Z_{\mathrm{obs}}$ during the fitting in the retrieval.

### 2.2.2    Brightness temperature

To simulate microwave brightness temperatures $T_B$, gaseous absorption by water vapor and oxygen is computed according to Rosenkranz (1998) and the absorption due to liquid water according to Liebe et al. (1993). The forward radiative transfer

calculation is then performed by integrating the radiation intensity along the vertical path up to an altitude of 30 km, neglecting the variation in optical depth due to scattering. As such, the $T_B$ measurements provide constraints on the liquid water path (LWP) of a given column.

### 2.2.3    Lidar attenuated backscatter

A publicly available code[1] for the calculation of lidar signals including multiple scattering is used to simulate the lidar at-

tenuated backscatter. In treating the multiple scattering, the code allows an explicit computation of higher scattering orders following an approach by Eloranta (1998), and a fast calculation using the photon variance-covariance method (Hogan, 2006, 2008). Once the relevant parameters are available, e.g. lidar set-up, extinction coefficient and droplet size profiles, lidar attenuated backscatter from below and within the cloud can be calculated.

### 2.3    The inversion scheme

### 2.3.1    State vector

The state vector refers to the collection of control parameters that we aim to optimize in order that our forward models match the observations. In this section we motivate the choice of the state vector. The details of how the state vector elements are implemented in the retrieval algorithm are discussed in Section 2.3.2. There are 12 elements in this vector: 7 for the cloud component, 4 for the drizzle component, and one element $\alpha_0'$ to compensate for a possible offset in the lidar signal due to

---

[1] http://www.met.reading.ac.uk/clouds/multiscatter/





imperfect calibration. $\alpha'_0$ is the extinction coefficient at the highest lidar range gate before cloud or drizzle is detected (see equations 21-25). Below we discuss the state vector elements for cloud and drizzle retrieval:

- Cloud: $[\nu, \hat{h}, W, N_{\mathrm{ad}}, ft_{\mathrm{cb}}, ft_{\mathrm{ct}}, p_{\mathrm{cb}}]$

  The first element $\nu$ specifies the shape of the cloud DSD (eq. 1) and it is set to be independent of height. The second and third elements determine $f(z)$ (eq. 9) and subsequently the LWC profile of the cloud. $N_{\mathrm{ad}}$ is the adiabatic number concentration (eq. 10 and 11). Since $f(z)$ is dependent on parameters normalized by the cloud depth, an estimate of cloud boundary altitudes becomes an important issue. What is especially critical is the model cloud base height because the LWC profile near the cloud base is derived from the adiabatic LWC at the cloud base and it increases almost linearly up to the height where the water content reaches its maximum value. A shift in the starting point (cloud base) would inevitably compromise the estimate for LWC in the lower and intermediate region within the cloud. This would then propagate to inaccuracies in the estimate of the other microphysical profiles. As far as the LWC model is concerned, cloud base and cloud top are the points where cloud droplets cease to exist, that is where liquid water content reaches precisely zero. Since real clouds are by nature 'fuzzy', our aim here is not to find an exact height where cloudiness starts or ends but rather to find an optimal height of cloud base and top to adopt in the $f(z)$ model that would lead to a good overall estimate of LWC across heights. The estimates of cloud boundary heights based on the radar reflectivity and the lidar attenuated backscatter data are somewhat coarse due to the limited range resolution and therefore more tuning is required. The variables $ft_{\mathrm{cb}}$ and $ft_{\mathrm{ct}}$ are included in the state vector to serve as tuning factors that allow the model cloud boundaries to be located at any spatial point within one range resolution from the lidar or radar-based estimates. This way, the model has the freedom to make necessary adjustments not only within the cloud but also at the boundaries.

  For cloud top, this tuning factor $ft_{\mathrm{ct}}$ is used to determine $z_{\mathrm{ct,opt}}$ and is fixed to a value in the range [-1,0]:

  $$z_{\mathrm{ct,opt}} = z_{\mathrm{ct}} + ft_{\mathrm{ct}}\Delta z, \tag{16}$$

  where $\Delta z$ is the range resolution of the radar and $z_{\mathrm{ct}}$ is the height of the radar range gate above the last radar detection.

  The treatment of the cloud base is less simple than the cloud top. The measured lidar signals appear to have suffered from spatial broadening around the cloud base: the attenuated backscatter increases towards the peak value more mildly than theoretically expected for clouds with sharp, well defined boundary, suggesting a somewhat gradual increase in extinction coefficient. This is true also for cases where there is no radar detection below cloud base. A likely cause for this is turbulence and entrainment that blurs the cloud boundary creating a transitional region seen in the lidar signal. This effect is comparatively minute in magnitude with respect to the maximum backscatter but would certainly affect the estimate of cloud base height and drizzle quantification. In the retrieval, we simulate this effect by a smoothing procedure. $ft_{\mathrm{cb}}$ has a similar function to $ft_{\mathrm{ct}}$ and $p_{\mathrm{cb}}$ indicates the level of the broadening (see Section 2.3.2).

- Drizzle: $[\nu_{\mathrm{dzl}}, \hat{h}_{\mathrm{dzl}}, W_{\mathrm{dzl}}, q]$ or $[\nu_{\mathrm{dzl}}, \alpha_{id}, \alpha_{\mathrm{cb}}, \alpha_{ic}]$

  There are two possible sets of elements that correspond to drizzle parameters. They share one common element, i.e. the shape parameter of drizzle DSD $\nu_{\mathrm{dzl}}$. The first set $[\nu_{\mathrm{dzl}}, \hat{h}_{\mathrm{dzl}}, W_{\mathrm{dzl}}, q]$ is used for column observations where drizzle is





not detected below cloud base. In such a case, even when accompanied by low radar reflectivity, small drizzle droplets can still be present within the cloud. On the other hand, the lack of a clear drizzle signature adversely limits the retrieval strategy so we resort to using the same vertical model as for the cloud (eq. 9). With this implementation, there are two pairs of $[\hat{h}, W]$ and hence two $f(z)$ in the system, one for the cloud and the other for the drizzle, independent of each other. $\mathrm{LWC_{ad}}$ is identical to that for the cloud, computed using the temperature and pressure at the model cloud base. For this reason, it is necessary to apply a scaling factor $q$ since drizzle and cloud LWC can be a few orders of magnitude apart, a range that is not covered by drizzle $f(z)$ alone.

The second set $[\nu_{\mathrm{dzl}}, \alpha_{id}, \alpha_{\mathrm{cb}}, \alpha_{ic}]$ is used for cases when there is drizzle detected below cloud base. $\alpha_{id}$ and $\alpha_{\mathrm{cb}}$ are extinction coefficients at the first radar gate (the lowest range gate with radar detection) and at the cloud base. At these two heights, $\mathrm{LWC_{ad}}$ is zero, allowing for an unambiguous drizzle retrieval. To construct a drizzle profile above the cloud base, one more control point is needed: $\alpha_{ic}$, which is the drizzle extinction coefficient at a certain height above the cloud base. Since the lidar signal suffers from strong attenuation inside the cloud, the height at which $\alpha_{ic}$ is retrieved is chosen to be 150 m above the cloud base.

### 2.3.2 Retrieval scenario

The retrieval is performed on a column-by-column basis and the algorithm is depicted schematically in Fig. 1. It starts with determining the cloud base heights by analysing the lidar return profile. There are two cloud base heights of relevance, one represents the actual cloud base ($z_{\mathrm{cb}}$) and the other is the model ('optimal') one used for LWC construction of the cloud ($z_{\mathrm{cb,opt}}$). We first find the height at which the lidar attenuated backscatter ($\beta$) is maximum, i.e. $z_{\mathrm{peak}}$. We then identify as $z_{\mathrm{cb}}$ the lowest range gate where $\beta$ rises by more than 50% to the next range gate and at the same time shows a continuous increase from there on up to $z_{\mathrm{peak}}$. The exact value of $z_{\mathrm{cb,opt}}$ is determined using the parameter $ft_{\mathrm{cb}}$ in the state vector and is restricted to around the midpoint between $z_{\mathrm{cb}}$ and $z_{\mathrm{peak}}$, such that:

$$z_{\mathrm{cb,opt}} = z_{\mathrm{min}} + ft_{\mathrm{cb}}(z_{\mathrm{max}} - z_{\mathrm{min}}). \tag{17}$$

$ft_{\mathrm{cb}}$ is constrained to the range of [0 ,1], $z_{\mathrm{max}} > z_{\mathrm{min}}$, $z_{\mathrm{min}}$ and $z_{\mathrm{max}}$ are larger than $z_{\mathrm{cb}}$ and smaller than $z_{\mathrm{peak}}$ respectively. By using $z_{\mathrm{cb,opt}}$, a model LWC for the cloud can be derived (eq. 8) where $\mathrm{LWC_{cld}}$ at the actual cloud base $z_{\mathrm{cb}}$, at cloud top and beyond the cloud boundaries is zero. Without further treatment, the use of this LWC profile will lead to a $\beta$ profile that shows a sharper turn just above the cloud base and a more rapid increase towards the maximum backscatter value than what is actually observed. To soften this behaviour, we smooth the model LWC profile in the region around $z_{\mathrm{cb,opt}}$ via an exponential moving average scheme. The distance $d$ between $z_{\mathrm{cb}}$ and $z_{\mathrm{cb,opt}}$ in the unit of lidar range resolution is used to derive the exponential weight of the smoothing $\exp(-p_{\mathrm{cb}}d)$ and the width of the smoothing window $\sigma$. This $\sigma$ is defined as the maximum number of consecutive range gates around $z_{\mathrm{cb}}$ that woud lead to a zero LWC at the actual cloud base. As the $\mathrm{LWC_{cld}}$ increases up to the peak value in an approximately linear fashion, the effect of the smoothing quickly diminishes with height. Therefore, the smoothing is performed only at $\sigma$ lidar range gates centered at $z_{\mathrm{cb,opt}}$ where the impact is significant.





In Fig. 2 we plot the profiles of the forward-modelled $\beta$ with and without LWC smoothing against observational data (circles). Without smoothing, the best-fit (forward-modelled) lidar backscatter is amplified strongly within 2 range gates (30m) before it gets attenuated (dashed line). In the height range between 1.20 and 1.26 km, the relative error at the signal peak is the smallest, leading to the good fit at this range gate at the expense of the much worse fit at the earlier range gates. The cloud base height inferred from the model $\beta$ appears to be an overestimate and at this height $\beta$ is underestimated by more than an order of magnitude. When LWC is smoothed around $z_{\mathrm{cb,opt}}$, the fit improves significantly as shown by the solid line.

The smoothed $\mathrm{LWC_{cld}}$, together with $N_{\mathrm{ad}}$ and $\nu_{\mathrm{cld}}$ provided in the state vector, are then used to compute the other microphysical profiles and the radar reflectivity of the cloud component $Z_{\mathrm{cld}}$. The difference between $Z_{\mathrm{cld}}$ and $Z_{\mathrm{obs}}$ is recorded as $Z_{\mathrm{excess}}$, such that:

$$Z_{\mathrm{excess}} = Z_{\mathrm{obs}} - Z_{\mathrm{cld}} \qquad \text{for } Z_{\mathrm{obs}} > Z_{\mathrm{cld}} \tag{18}$$
$$= 0.0 \qquad \text{for } Z_{\mathrm{obs}} \leqslant Z_{\mathrm{cld}}. \tag{19}$$

To enforce some level of spatial continuity for the drizzle above $z_{\mathrm{cb}}$, the resulting $Z_{\mathrm{excess}}$ is smoothed. For simplicity, we apply a simple moving average to $Z_{\mathrm{excess}}$ values within cloud boundaries using three range gates as the smoothing width to result in $Z_{\mathrm{dzl}}$. At this point, we can differentiate between two cases:

– No drizzle detection below cloud base (case I)

Since there are no obvious signs of drizzle in this case, there is a possibility that the amount of drizzle is simply too low to retrieve (e.g. lower than detection limit, lower than the measurement uncertainties). If $Z_{\mathrm{dzl}}$ is found to be positive at only 1 or 2 radar range gates we classify the case as non drizzling. Consequently, the drizzle properties are all set to zero at all heights and the algorithm proceeds to compute the observables considering only contributions from the cloud. If, on the other hand, there is sufficient detection for drizzle, $\mathrm{LWC_{dzl}}$ and the other drizzle properties are derived from the parameters in the state vector in a similar way to those for the cloud. Drizzle base and top height are set to the closest radar range gate beyond the first and last detected $Z_{\mathrm{dzl}}$, respectively, where $Z_{\mathrm{dzl}} = 0$.

– Drizzle detection below cloud base (case II)

The drizzle retrieval is based on the parametrization of $r_e$ vertical profile above (eq. 13) and below (eq. 14) the cloud base. The $r_e$ profile is split into two functional forms to account for the different expected behaviours of drizzle droplets across heights. To aid this parametrization, three values of $r_e$ at three different heights are needed: within the cloud, at the cloud base and below the cloud base. These values are produced via $Z_{\mathrm{dzl}}$ and extinction coefficients $\alpha_{id}, \alpha_{\mathrm{cb}}, \alpha_{ic}$ given in the state vector (see eq. 20) by combining equations 3, 5 and 7:

$$r_{e,\mathrm{dzl}}^4 = \frac{\pi Z_{\mathrm{dzl}}}{32\alpha_{\mathrm{dzl}}} \frac{(\nu_{\mathrm{dzl}}+2)^3}{(\nu_{\mathrm{dzl}}+3)(\nu_{\mathrm{dzl}}+4)(\nu_{\mathrm{dzl}}+5))}. \tag{20}$$

This way, we make the most of the constraints provided by the lidar since extinction coefficient strongly influences the lidar return. Below the cloud, $Z_{\mathrm{dzl}}$ is equal to $Z_{\mathrm{obs}}$, which allows drizzle properties to be retrieved without much ambiguity. After the drizzle effective radius is profiled, one can easily derive the other microphysical properties. As in case I, drizzle base and top are set to the closest radar range gate beyond the first and last $Z_{\mathrm{dzl}}$, respectively.





Following the above scheme, both the cloud and drizzle properties can be computed for a given state vector. There is, however, no guarantee that these properties are a sensible representation of the system in question. To mitigate this problem we impose several physical constraints that act as filters for the state vector:

- We apply a droplet size threshold to separate the cloud and drizzle regime. It has been shown that there exists a critical effective radius between 12-14 microns, above which coalescence increases and drizzle forms very rapidly (Rosenfeld et al. (2012) and references therein). Based on this, we adopt 13 microns as the separation threshold, which means that at any altitude, cloud $r_e$ has to be smaller than 13 microns and drizzle $r_e$ cannot be less than that.

- The radar reflectivity due to drizzle must not be higher than the cloud reflectivity near the cloud top. The cloud top region is critical for the cloud-drizzle separation because this region is where the drizzle starts to form and where the difference between cloud and drizzle droplet size is minimal. Since the cloud LWC is highest near the cloud top, it is likely that the cloud number concentration and therefore cloud reflectivity will be dominant here. From this it also follows that the location of the maximum radar reflectivity near the cloud top is an indicator of the location of the $LWC_{cld}$ peak.

- Drizzle effective radius achieves its maximum value at the cloud base. This follows from the scenario that drizzle droplets grow as the they fall through the cloud layers via accretion and evaporate after they leave the cloud, thereby reducing their size.

- Drizzle effective radius must not be larger than 250 microns due to the use of Rayleigh approximation on which the radar forward model is based. For a 35-GHz radar, the validity of the approximation sets an upper limit of droplet radius at about 280 microns. The 250 micron upper limit is imposed as a safeguard and is more of a technical limitation than a physical one. For the selection of drizzling clouds in our study here, this is not a concern.

If the cloud and drizzle properties meet all the physical requirements, the algorithm proceeds to computing the observables. If, on the other hand, the cloud and drizzle properties do not comply with all of the constraints above, the state vector from which they are calculated will be discarded altogether by setting the cost function (eq. 26) to a high value. The optimization routine then continues to repeat the computation from the beginning using a different state vector. One exception is for the third constraint for case I where we penalize the cost function instead when the drizzle effective radius does not monotonically decrease with height (Section 2.3.3).

The total radar reflectivity is computed by summing up $Z_{cld}$ and $Z_{dzl}$ and applying the liquid attenuation factor. To calculate the brightness temperature, additional information on the pressure, temperature and humidity (from a numerical forecast model or a radiosonde) is required by the forward model. For lidar, the profile of extinction coefficients for the drizzle-free altitudes between the ground and the cloud is needed to provide an estimate for lidar calibration offset and to construct a complete attenuated backscatter profile. At these altitudes, scattering due to air molecules and aerosol particles are expected to prevail. While it is straightforward to approximate the extinction coefficient due to air molecules $\alpha_m$ from the temperature and pressure profile, the aerosol extinction coefficient $\alpha_a$ is largely unknown. Since multiple scattering does not play an important role in this region below the cloud/drizzle base, we use Klett inversion for a two-component atmosphere (Klett, 1981; Kovalev, 1995)





to infer $\alpha_a$ from the observed $\beta$ profile, such that for $z < z_0$:

$$\alpha'(z) = \left[ \frac{\left( \frac{P'(z)z^2}{P'(z_0)z_0^2} \right)}{\frac{1}{\alpha_0'} + 2\int_z^{z_0} \frac{P'(z')z'^2}{P'(z_0)z_0^2} dz'} \right] \tag{21}$$

where

$$\alpha'(z) = \alpha_a(z) + S_a\beta_m(z), \tag{22}$$

$$P'(z) = S_a P(z) \exp\left( 2\int_0^z \left( \alpha_m(z') + S_a\beta_m(z') \right) dz' \right). \tag{23}$$

The zero subscript refers to the Klett reference point with $z_0$ is set to $\min(z_{cb}, z_{db})$ and $\alpha'(z_0)$ is equivalent to $\alpha_0'$, one of the state vector elements. $S_a$ is the extinction-to-backscatter ratio for aerosol. Given that the lidar operates at 355 nm, we adopt $S_a = 50$ sr, a representative value for aerosol particles. $\beta_m(z) = \alpha_m(z)/S_m$ is the attenuated backscatter due to molecular scattering and is calculated assuming $S_m = 8\pi/3$. $P(z)$ is the attenuated backscatter power as a function of height and is defined as:

$$P(z) = \frac{C_{ldr}(\beta_z(z) + \beta_m(z))}{z^2 \exp\left( 2\int_0^z (\alpha_a(z') + \alpha_m(z')) dz' \right)}. \tag{24}$$

Using the equations above, the $\alpha_0'$ value in the state vector, and the fact that lidar calibration factor $C_{ldr}$ cancels out in eq. 21, the $\alpha_a$ vertical profile below the reference point can be derived. Together with the extinction coefficient of the drizzle and the cloud, $\alpha_a(z)$ is then used as input for the lidar forward model to construct the complete attenuated backscatter profile below and within the cloud/drizzle. The lidar calibration factor $C_{ldr}$ can now be computed at each range gate in the cloud- and drizzle-free region:

$$C_{ldr}(z) = \frac{S_a\beta_{obs}(z) \exp(2\int_0^z (\alpha_a(z') + \alpha_m(z')) dz')}{\alpha'(z)}. \tag{25}$$

Since $C_{ldr}$ serves to compensate for a systematic offset due to inaccurate calibration, its values are expected to be approximately constant with heights. We have confirmed that the values are very similar across heights to within about 2%. Finally, we multiply the forward-modelled $\beta$ profile by the median value of $C_{ldr}(z)$ for comparison with the observed $\beta$.

### 2.3.3 Finding the optimal solution

This retrieval procedure attempts to solve the inverse problem of deriving cloud and drizzle properties from observations by minimizing the cost function

$$cf = [y - F(x)]^T S_y^{-1}[y - F(x)] \tag{26}$$

to arrive at the optimal solution for the state vector $x$. $y$ is the measurement vector defined as:

$$y = [T_{B,obs,1}, ... T_{B,obs,nf}, \beta_{obs,1}, ..., \beta_{obs,nl}, Z_{obs,1}, ..., Z_{obs,nr}], \tag{27}$$




with $nf, nl, nr$ represents the number of MWR frequency channels (14), the number of lidar range gates, and the number of radar range gates with detection, respectively. $F(x)$ is the vector of forward-modelled observables, with the same composition as $y$. For a $T_B$ measurement at a frequency $i$ with an uncertainty $\sigma_{T_B,i}$, the diagonal element $(i,i)$ of the measurement covariance matrix $S_y$ is $\sigma^2_{T_B,i}$. All non-diagonal elements (cross-channel or cross-instrument elements) are set to zero assuming no

correlation. The elements $(m,n)$ of matrix $S_y$ corresponding to radar and lidar data are calculated according to

$$S_{y,m,n} = E\left([y_m - E(y_m)][y_n - E(y_n)]\right), \tag{28}$$

which results in:

$$
\begin{aligned}
S_{y,m,n} &= \sigma^2_C y_m y_n & \text{for } m \neq n \tag{29}\\
&= \sigma^2_C y_m^2 + \sigma^2_{y_m} & \text{for } m = n \tag{30}
\end{aligned}
$$

where $\sigma_{y_m}$ and $\sigma_C$ are the random uncertainties of the measured signal and the instrument calibration, respectively. $\sigma_C$ is set to be small (comparable to the desired fit accuracy for $Z$ and $\beta$.

For case I, a penalty term is added to the cost function to bias $r_{e,\text{dzl}}$ toward the desired profile. Along the drizzle profile, it is checked whether $r_{e,\text{dzl}}$ is larger than the one directly below it. Since the determining factor of drizzle is the radar reflectivity, the penalty is applied to the radar part of the cost function such that every single violation would add the radar term to the

original cost function.

The cost function (combined with the penalty function for case I) is minimized using Differential Evolution (DE), a global stochastic optimization technique similar to population-based optimization routines (Storn and Price, 1997). DE does not require gradient information, which is an advantage given the complexity and the non-linearity of the cost function. It is designed to deliver robust results and a fast convergence while maintaining a small number of control variables. This minimization algo-

rithm begins with a population of state vectors that constitute a generation. The vector values are chosen to cover the allowed parameter space and the population is then updated with each generation. For each member vector, a new vector is created through mutation and parameter mixing/crossover to replace the old one if it results in a smaller cost function value. Otherwise, the old member vector is retained to be part of the subsequent generation population. This mutation-crossover scheme, along with the strategy to start with a a set of vectors, instead of a single initial vector, make it less likely for the algorithm to

get trapped in local minima. To use the algorithm, initial guesses for the state vector values are not needed, but the lower and upper limits for each state vector parameter are required.

DE comes in several variants, which differ in the way the mutation and crossover are done. Here we choose the DE/best/1/bin variant with a population size NP=10, a mutation factor F that randomly changes between 0 and 1.9 on a generation-by-generation basis, and a crossover constant CR=0.8 (Storn and Price, 1997). For the retrieval, we use the numerical implemen-

tation of DE provided within the Python-based environment for scientific computing SciPy[2], where the stop conditions are specified by the tolerance (0.01) and the maximum number of generations (150).

The minimization of the cost function is performed over bounded state vector values, from which a physically-sensible solution should be found. Unless stated otherwise, the lower and upper limits of state vector values that we use in this work are

---

[2]http://www.scipy.org





listed in Table 1. The shape parameter for the cloud DSD is expected to vary between 2 and 10 (Miles et al., 2000; Gonçalves et al., 2008), depending on for example airmass and location (marine or continental). For drizzle DSD, an exponential fit ($\nu = 1$) is found to be a good approximation (Wood, 2005b). Here we allow $\nu$ to vary within a wide range. From our investigation (Section 3) it appears that constraining $\nu$ to a fixed value when the radar calibration accuracy is unknown can potentially create

a significant bias in the retrieval products. The limits for $\hat{h}$ and $W$ cover the subadiabatic range of LWP that is viewed to be common (Boers et al., 2006). Since small drizzle droplets present minimal effects on $\beta$, the extinction coefficient of drizzle at the cloud base $\alpha_{\mathrm{cb}}$ is constrained to be comparable to the air extinction coefficient. The value of $\alpha_{\mathrm{id}}$ is expressed relative to the drizzle extinction coefficient at the cloud base and $\alpha_{\mathrm{ic}}$ is determined relative to the value of the cloud extinction coefficient value at the same height. $\alpha_0'$ is given a large range because it is rather sensitive to a small change in the retrieved lidar offset.

The uncertainties for the optimal solution are computed using Monte Carlo realizations that were generated by perturbing the observations. Each random realization of the observations is drawn from a Gaussian distribution centered on the measurements with the dispersion taken from the measurement (random) errors. The retrieval procedure is then performed on all realizations resulting in a set of solutions. The RMS difference with the optimal solution is calculated to represent the uncertainties of the retrieval. For each column observation, we create ten realizations which should provide a conservative estimate of the random

uncertainty. Systematic uncertainties due to inaccurate radar calibration are not included in the Monte-Carlo error estimate. Assuming that the calibration offset can be under- or overestimated by up to a factor of two (3 dBZ), the resulting systematic errors on the retrieval products are found to be larger than the random uncertainties (see Section 3).

## 3    Test using synthetic data

Before applying the technique to real observational data we test it on a set of synthetic signals generated from large-eddy

simulation (LES) results. Similar to the work described in Donovan et al. (2015), the simulation set-up is based on output from the Dutch Atmospheric LES model (DALES) (Heus et al., 2010) for conditions corresponding to the FIRE campaign (Curry et al., 2000). Given the LWC from the LES, the DSD is assumed to be a monomodal gamma distribution, i.e. drizzle droplets are not present. The shape parameter and the number concentration along the vertical column are externally imposed and they are largely constant. ECSIM (Voors et al., 2007; Donovan et al., 2015) was used to generate the radar, lidar and MWR signals.

Applying the algorithm to these signals serves primarily as a sanity check for the retrieval code, to verify the forward models and the assumption on the vertical shape of LWC.

The synthetic signals are simulated for a zenith-pointing 32 GHz radar, a lidar operating at 353 nm, and a MWR with 14 frequency channels between 20-60 GHz to mimic the instruments used in the ACCEPT campaign (see Section 4). The radar and lidar signals are sampled at a fine spatial resolution: 2.5 m vertically and 25 m horizontally. To mimic real observations we

degrade the vertical resolution of both the radar and lidar to 22.5 m. Along the horizontal axis (corresponding to the time axis), we lower the resolution to 150m by averaging radar reflectivity and lidar attenuated backscatter data at each range gate, and averaging the brightness temperatures at each frequency channel. The standard deviation of the mean serves as the uncertainty. The atmosphere below the cloud is rather static, making the standard deviation of $\beta$ in this region unrealistically low. This



condition virtually assigns a lot of weight to the part below the cloud in the fitting process, which leads to inaccurate retrieval. For this reason, we set the noise floor for the $\beta$ profile to 1% below the cloud base and 5% above the cloud base. Similarly for $Z$ and $T_B$, the minimum relative error is set to 0.03 and 0.01, respectively. It is these simulated measurements with adjusted resolution, together with the uncertainties, that are fed to the retrieval code.

Fig. 3 shows the input synthetic signals as compared to the signals recovered by the retrieval. Apart from the lower edge of the cloud at horizontal distances $< 7.5$, $Z$ is generally well reproduced. The lidar signal is also recovered despite the noise. It is fitted up to 300 m into the cloud, after which the noise prevails. The histograms of the reflectivity and the attenuated backscatter residuals (truth - retrieval) are displayed in panels 3e and 3f. Most of the residuals are relatively small, the peaks of the histograms are centered at zero and the distributions are quite symmetric with no particularly strong tendency towards
positive or negative values. $T_B$, averaged over distance, at each frequency channel coincides well with the data. Average $T_B$ fluctuations over time are small: less than 4% as shown in Fig. 3f. The Root-Mean-Square Deviation (RMSD) between the data and the retrieval is also very low, i.e. less than 1%, suggesting a good match between the two. The maximum rms is found at 31.4 GHz, where the extinction due to liquid water dominates the microwave signal.

Fig. 4 displays the true microphysical and optical properties in comparison with the retrieved ones. The structures in the
LWC, $r_e$ and $\alpha$ are mostly reproduced. Since the retrieval is performed on a column-by-column basis the retrieval is not entirely smooth along the horizontal axis and this effect is particularly visible in $N$. The mild vertical structure in the true droplet concentration is not reproduced due to the model assumption of constant $N$. From the histograms, it can be seen that generally the LWC and the extinction coefficient are retrieved more accurately than the effective radius and the number concentration. The retrieved $r_e$ and $N$ tend to be higher and lower than the truth, respectively, by a few percents (see also
Fig. 5 and Table 2). The distribution of $\Delta N$ appears less Gaussian than those for the other microphysical properties due to the column gradient in the true $N$ that is not matched by the retrieval assumption.

Fig. 5 displays the vertically collapsed version of Fig. 4. LWC and $\alpha$ are integrated into LWP and optical depth, respectively. $N$ and $r_e$ are vertically averaged, with the latter weighted by $\alpha$. The error bars represent the random measurement error from the Monte Carlo realizations; there is no systematic error due to the radar calibration. The fluctuations of the variables along the
horizontal axis are easily reproduced with very little bias, which is mostly seen in $r_e$ and $N$, as shown before by the histograms in Fig. 4. The mean values of the LWP, $r_e$, optical depth and the number concentration, averaged over the horizontal axis, and the deviation from the truth are given in Table 2.

The true shape parameter is not strictly constant along the vertical direction; it is mostly close to 6, and decreasing to around 2 at the cloud base or cloud top. The retrieval is performed with $\nu = 5.5$ and with the radar calibration factor fixed to 1 to
match the true values. The lidar calibration factor is retrieved on average with a 5% accuracy. For comparison purposes, we also include in the last column of Table 2 the run where the shape parameter $\nu$ is free within a fixed range, i.e. between 2 and 10. The result is that the noise of the retrieved products becomes higher but there is very little systematic offset. The optimized $\nu$ is found to have a mean of 5.98 (RMSD=2.01), very close to the true $\nu$. By comparing the last two columns in Table 2, it is apparent that when $\nu$ is not fixed, the RMSD increases significantly due to the large column-to-column fluctuation but the



mean values are hardly affected. The extinction coefficient is found to be relatively stable against the variation in $\nu$, possibly because its retrieval is largely dependent on the $\beta$ profile.

We also investigate the effect of under- or overestimating the radar calibration offset. For this purpose, we apply a shift of $\pm 3$ dBZ (a factor of two) to the forward-modelled $Z$ and perform the retrieval with $\nu$ bounded between 2 and 10. When the offset is underestimated (forward modelled $Z$ is multiplied by 0.5, or $C_r = 0.5$), LWC and $r_e$ are overestimated by 10-15% while $\alpha$ generally becomes lower by a few percent, and vice versa when $C_r = 2$. The retrieved number concentration tends to fluctuate and is on average 15% higher than the mean truth for $C_r = 2$. The relatively mild systematic impact of doubling or halving the radar calibration offset is possibly due to the fact that the shape parameter is allowed to vary within a certain range; the true shape parameter is not recovered in both cases of $C_r$. The magnitude of the systematic difference between the retrieval products and the truth increases when $\nu$ is fixed to the true value, especially for the number concentration where the mean retrieved $N$ becomes 36% lower than the truth.

What is demonstrated with this exercise is that the forward models are able to reproduce the radar, lidar and MWR signals and that the LWC parametrization that we use for the cloud indeed provides a realistic description of LWC vertical structure. Given accurate instrument calibration, the systematic mismatch between the retrieval and the truth is found to be very small for this test case, both when $\nu$ is fixed to approximately the true value or when it is optimized. From all four retrieval products, the largest mean offset from the truth is found for the number concentration $N$ at less than 5%.

## 4 Application to ground-based observations

The observational data was collected during the ACCEPT campaign that took place in October and November 2014 in Cabauw, The Netherlands (see Myagkov et al. (2016) and Pfitzenmaier et al. (submitted) for more information about the measurement campaign). We use the data acquired from three co-located instruments:

- A zenith-pointing MIRA-35 radar

  It is a Ka-band cloud radar with Doppler capabilities. The signal was recorded with a spatial resolution of about 30 m.

- A UV lidar (Leosphere ALS-450) operating at 355 nm

  The attenuated backscatter measurements are available every 30 sec with a vertical resolution of 15 m.

- A microwave radiometer (MWR) HATPRO

  The brightness temperature was measured at 14 frequency channels: the first seven between 20-35 GHz and the other seven between 50-60 GHz. The temporal resolution is 1 sec with regular gaps due to automatic calibration periods.

For the inversion procedure, we use the calibrated radar reflectivity factor, as well as model forecast of temperature and humidity delivered in the Cloudnet categorization product (Illingworth et al., 2007). The calibrated reflectivity here is already corrected for gas attenuation and has the same temporal resolution as the lidar although their time stamps do not exactly coincide.



The retrieval is performed on a column-by-column basis with a time interval of 30 sec. For each 1D column, a set of radar, lidar and MWR data was created by first finding the lidar and radar profiles that are less than 15 sec apart. The corresponding $T_B$ profile was computed by averaging $T_B$ measurements within 15 sec of the average time stamp of the radar and lidar. The standard deviation of the mean was then adopted as the measurement error. Since the full overlap distance of the lidar is

expected to be around 100-200m, column profiles with radar detections down to $< 200$m were not retrieved. There are gaps in the observations where a complete data set for the three instruments is not available, e.g. breaks in the MWR data stream during instrument calibration periods.

We selected two periods with a total time of approximately 4 hours on October 25 and 26, when one layer of liquid water cloud is present. The scene includes clouds with clear drizzle/precipitation events and also clouds without obvious signature

of drizzle below the cloud base, suitable for the dual retrieval mode (case I and case II). The cloud top is located between 1400 and 1500 m, with the cloud thickness varying between 200-400 m. The cloud base height (as determined in the retrieval) fluctuates between 1050 and 1250m during the two periods. Despite low reflectivity values, virga/drizzle is observed below the cloud base for the majority of the time with its maximum occuring on Oct 26. The extent of the drizzle below the cloud is variable, with a depth of up to 600 m.

The observed signals and their recovery in the retrieval is shown in Fig. 6 for each instrument. In general, the reflectivity within the cloud increases with height indicating particle condensational growth. For the most part, the radar reflectivity is not higher than -28 dBZ. On Oct 26 at around 3.8hr, $Z$ is maximum at -12 dBZ. In the retrieval, cloud and drizzle contributions to the total reflectivities are separated and these are shown in panels (c) and (d). Below the cloud base, $Z$ belongs only to the drizzle. Above the cloud base, the reflectivity of the cloud increases with height and peaks close to the cloud top.

Conversely, drizzle reflectivity increases downwards from the top and reaches maximum in the cloud base region before decreasing again towards the drizzle base. It follows that within the cloud, the cloud reflectivity dominates towards the cloud top while drizzle dominates near the bottom. In almost all profiles where no virga is visible below the cloud base, the retrieval algorithm finds drizzle to be present within the cloud although with small reflectivities. This is usually caused by the significant excess reflectivity near the cloud base that cannot be attributed to the cloud component.

Panels (e) and (f) compare the observed and retrieved lidar attenuated backscatter. The fitting of the $\beta$ profile starts from an altitude between 200 m (from the ground) and the drizzle base, and continues up to 200 m above the cloud base. Several lidar $\beta$ profiles show double backscatter peaks that we deem unsuitable for the algorithm, in which case the retrieval is not performed resulting in white gaps in the time-height map. We found 25 such column profiles, corresponding to 5% of the available data. It can be seen that the drizzle below the cloud base remains transparent and undetected by the lidar, which is exactly a condition

assumed in the retrieval. The histograms of the residual signals are similar to those obtained for the LES exercise (Fig. 3e and 3f): centered at zero and largely symmetric. Compared to Fig. 3e, the $\Delta Z$ distribution here is narrower because a part of the Z residual is attributed to drizzle. The mean brightness temperature at each frequency channel is shown in Fig. 6i. The observations show little variation over time (less than 8%). On average, the fit to the observed $T_B$ is reasonably good with a RMSD of 5% or less, as indicated by the red line. The largest variation or difference is seen around 30 GHz, where the cloud

contribution to the microwave extinction spectrum is significant.



The retrieved microphysical and optical properties for both cloud and drizzle are shown in Fig. 7. Cloud and drizzle LWC in panels (a) and (b) show a similar time-height distribution to the respective reflectivity field. Cloud LWC increases with height until the peak is reached close to the cloud top, while most water in drizzle is found at a lower altitude. The highest LWC is found at the time of maximum observed reflectivity. Drizzle water content is generally 2 orders of magnitude smaller than

the cloud LWC. The average of $LWC_{cld}$ and $LWC_{dzl}$ maximum values are $1.9 \times 10^{-1}$ and $1.2 \times 10^{-3}$ g/m$^3$, respectively. The temporal variations of the cloud and drizzle LWP are positively correlated in time, as was also found iby Fielding et al. (2015).

The effective radius of the cloud droplets (panel c) is found to be well below the threshold values of 13 microns. As the cloud droplets grow via condensation, their size increases with height to a peak value of 5.1 microns, on average. Drizzle effective radius (panel d), on the other hand, increases towards the cloud base as imposed by the parametrization. At the cloud base, the

mean drizzle effective radius is found to be $\sim 22$ microns. During the intense drizzle period between 3.8-4.0 hr, it can be as high as 60 microns.

The extinction coefficients of the cloud and drizzle (panels e and f) are mainly determined from the observed lidar attenuated backscatter. The extinction coefficient of drizzle is smallest below the cloud base, in accordance with the relatively low observed $\beta$ in this region. It increases with height and peaks within the cloud but it is still orders of magnitude smaller than the cloud.

The mean maximum $\alpha_{cld}$ and $\alpha_{dzl}$ are 0.06 and $1.2 \times 10^{-4}$ m$^{-3}$, respectively. The number concentration of the cloud (panel g) droplets shows a somewhat high and rapid fluctuation. From our LES exercise (Section 3), we learned that setting $\nu_{cld}$ as a free parameter can indeed cause this, but we also expect that the fluctuation averages out to a minimally biased mean value. The cloud number concentration averages to about 549 cm$^{-3}$. The drizzle droplet concentration (panel h) has a mean of $\sim 0.06$ cm$^{-3}$, consistent with the in-situ measurements of marine-stratocumulus in the MASE-II experiment (Lu et al., 2009).

In Fig. 8 we show the mean vertical profiles of the radar reflectivity and the derived microphysical properties of the cloud and drizzle. These profiles are constructed from averaging the retrieved profiles between 3.8 and 4.0 hour UTC when there is significant drizzle. In all but the lowermost parts of the cloud, drizzle gives a very small contribution to the total water content. Close to the cloud top, the effective radius of the drizzle droplets are found to be around 30 microns for this particular time segment. At around the same height, the cloud droplets reach their maximum size of $\sim 6$ microns with the cloud liquid water

content and reflectivity dominating over the drizzle. As the drizzle droplets grow exponentially towards the cloud base via coalescense, its reflectivity increases, matching the cloud reflectivity about halfway through the cloud. Near the cloud base, drizzle reflectivity becomes dominant due to the much larger size of the drizzle droplets compared to the cloud droplets. Inside the cloud layers, this behaviour of cloud and drizzle reflectivities is found to be quite typical over the observation period. From the cloud base towards the drizzle base, $Z_{dzl}$ decreases monotonically, and so does the $LWC_{dzl}$. Once drizzle drops escape the

cloud, they are expected to evaporate below cloud base and shrink as they fall through the air, in accordance with the gradient seen in the $r_{e,dzl}$ profile. The jagged feature in the $r_{e,dzl}$ profile below the cloud base is an artifact of the profile averaging caused by the variable drizzle base height during the 12-minute period.

Lastly, the number concentration of the drizzle is 3-4 orders of magnitude smaller than the cloud at all heights above the cloud base. Unlike the cloud number concentration, $N_{dzl}$ shows variation in height. The highest density of drizzle is found

approximately where its LWC is highest, that is within the cloud. From this point towards the cloud base, the drizzle number





density keeps decreasing as $r_{e,\mathrm{dzl}}$ rises sharply, which could be due to to the accretion of smaller drizzle droplets by the bigger ones to form even larger droplets. Below the cloud base, some droplets experience complete evaporation, depleting the number density of drizzle as it approaches the ground.

## 5  Comparisons with Other Retrieval Methods

To assess the results presented in Section 4, we perform comparisons with three independent retrieval methods, applied to the ACCEPT dataset within the same period. The three retrieval techniques offer a tool to evaluate our retrieval below and above the cloud base separately. Below the cloud base, the drizzle comparison is made with the readily-available results from the method of O'Connor et al. (2005) as part of the Cloudnet algorithm package. Above the cloud base, the cloud properties are retrieved using the depolarization lidar technique developed by Donovan et al. (2015). The amount of drizzle within the cloud is derived from radar doppler spectra analysis, as described in Kollias et al. (2011a, b). In Appendix A, we present the application of this technique to the ACCEPT data, following the implementation in Luke and Kollias (2013); the comparison with our retrieval is discussed in Section 5.2.

### 5.1  Cloud properties above the cloud base

A depolarization lidar-based (DL) method (Donovan et al., 2015) was applied to the lidar dataset used in our retrieval to derive the cloud properties. While our retrieval method utilises only the total attenuated backscatter, the DL method exploits the parallel and perpendicular polarization components of the received signal to infer the cloud extinction coefficient and droplet size.

Fig. 9 displays the time series of the cloud properties at a specific, arbitrary height (chosen to be 100 m) above the cloud base, as derived from the DL (blue) and our (black, red) retrieval methods. The DL method determines the cloud base height that is then used as a height reference for the subsequent retrieval of the extinction coefficient and effective radius, from which the LWC and $N$ are then derived. The retrieval is performed with a temporal resolution of 180 sec. For a fair comparison, we interpolate our retrieved profiles to 100 m above the cloud base as well. Since our retrieval has a higher temporal resolution (30 sec), we use our own cloud base height estimate ($z_{\mathrm{cb}}$) as a reference. It is therefore imperative to first make sure that the cloud base height estimates from the two methods match. The bottom panel of Fig. 9 shows that this is indeed the case.

We average our results over the 180-sec time interval to match the time stamps and the temporal resolution of the DL method. The time-averaged products (red line in Fig. 9) are then compared with the results of the DL method. The extinction coefficients retrieved by both methods are very similar; we find that the mean difference and the RMSD amount to -5% and 11%, respectively. The fractional quantities here and in the following are produced using the mean of the DL results as a reference. The effective radii do not compare as well, with a mean difference of -10% and a RMSD of 21%. Our effective radius is highest at around 3.88 hr, coinciding with a high radar reflectivity situated in the middle of an intense drizzle episode. This is not picked out by the DL retrieval. The difference in $r_e$ is mostly visible between 17.0-17.4 hr and 4.8-5.4hr. The

mismatch does not necessarily correlate with the drizzle quantity and can amount to up to 1.5 microns but is still within the expected DL uncertainty range.

Our retrieved LWC is on average lower by about 15% with a RMSD of 26%. The DL number concentration is derived assuming that the DSD follows a single-mode gamma distribution (eq. 1) with a shape parameter $\nu = 6$. Uncertainties of $N$ are propagated from the errors of $r_e$ and $\alpha$, taking into account a range of shape parameter values between 4 and 10 which is typical for liquid water clouds. Our number concentration fluctuates rather strongly with time, contributing to the large RMSD (45%). The shape parameter in the retrieval procedure is set as a free parameter to minimize biases and as discussed in Sect 3, this can lead to a strong fluctuation in the number density. In comparison to the DL method, our mean $N$ is higher by about 15%. Given that the DL uncertainties of $\alpha$ are between 15-20%, and that the large fraction of the DL-retrieved $r_e$, LWC and $N$ have $\sim 50\%$ errors, the differences between the two methods are well within the range of uncertainties.

## 5.2 Drizzle reflectivity above the cloud base

The qualitative comparison between our and the Doppler retrieval results is shown in Fig. 10 for the drizzle (a), cloud (b) and the total (c) reflectivities. Each circle represents a time-height pixel for which both methods are applicable; the number of time-height pixels that can be retrieved using only one of the methods, and hence are not used in the comparison, is less than 8%. In general, the distribution of the circles is consistent with the one-to-one line. The spread is higher for lower $Z$, indicating the higher fraction of noise for low-$Z$ retrieval. The larger values of our ('retrieved') total Z in (c) compared to the Doppler counterpart can be attributed to the absence of attenuation correction in the Doppler spectra. On average, there is a difference of 0.9 dBZ between the two sets of total $Z$. This difference has very little effect on the trend that we see in the scatter plots (a) and (b), in which the circles are color coded according to their relative heights within the cloud. We divide the cloud into three horizontal parts: (i) one quarter into the cloud (cloud base region: red), (ii) the top quarter (cloud top region: green), and (iii) the middle region in between (blue). The insets show the same scatter plot without the blue points to highlight the division between the cloud base and cloud top regions.

The correlation between our and Doppler $Z$ varies across altitudes within the cloud. Close to the cloud base there is a clear tendency that our retrieved drizzle reflectivities are larger than those from the Doppler analysis. The cluster of red circles in (a) is almost exclusively located to the left of the one-to-one line. Consequently, our cloud reflectivities become smaller than the Doppler $Z_{\mathrm{cld}}$, as seen in (b). The primary cause for this stems from the assumptions used in both methods. In our retrieval, the cloud LWC, effective radius and thus reflectivity near the cloud base is assumed to decrease downwards until it reaches zero at the cloud base. This way, drizzle can gradually maximize its share of the total reflectivity, guaranteeing the continuity of the drizzle reflectivity when crossing the cloud base. Such a restriction in the vertical profile is not in place for the Doppler retrieval and the applicability of the method is limited to situations where drizzle is not dominant.

It should be noted that the accuracy of the Doppler retrieval is crucially determined by the shape of the spectra. The choice of the time interval during which the individual spectra are averaged is known to play a role in determining the shape of the composite spectra (Giangrande et al., 2001). Here we fix the time window to 30 seconds to allow for a one-to-one comparison with our retrieval results. High turbulence is shown to cause an underestimation of drizzle reflectivity, as derived through





spectral decomposition, by up to 10-15 dBZ (Luke and Kollias, 2013). This means that even if drizzle indeed dominates the cloud, turbulence can smear the spectrum in such a way that it appears cloud-dominated. A compensation for this effect is thus critical in determining the correct drizzle fraction. In the scatter plots above, the presented reflectivity values have been turbulence-corrected according to Luke and Kollias (2013) who estimate the correction factor as a function of spectral broaden-

ing, determined from their extensive dataset of marine stratocumulus clouds. Ideally, turbulence-corrected drizzle reflectivities just below and just above the cloud base should have similar values, which was indeed the case in Luke and Kollias (2013). After applying their procedure to the ACCEPT dataset, however, we find that the drizzle reflectivities above the cloud base is lower by several dBZ (the mean is 4 dBZ lower) than those just below, as shown in Fig. 11. This could be the result of the uncertainties in the estimate of the correction factor, or an artifact of the spectrum averaging process. We do not attempt to

formulate the correction factor necessary to eliminate this reflectivity gap due to the insufficient amount of data to provide a statistically significant estimate.

The best agreement between the two methods is found for the middle part of the clouds. If a shift of a few dBZ is introduced to $Z_{\mathrm{dzl}}$ above the cloud base to correct for the reflectivity gap indicated in Fig. 11, then the agreement would improve in the cloud base and mid-cloud regions, but would worsen in the cloud top region. Overall, the correlation coefficient assessed by including all points is 0.38 for drizzle and 0.75 for the cloud. When we examine only the mid-cloud reflectivity distribution,

the correlation coefficient improves to 0.54 and 0.87 for drizzle and cloud, respectively. The cloud boundaries are problematic areas for the comparisons due to their transitional nature, lower reflectivities and therefore high uncertainties. However, despite the distinct retrieval procedures, different sources of information and different assumptions, the two methods show reasonable agreement in quantifying the amount of drizzle within the cloud.

### 5.3   Drizzle LWP below the cloud base

Drizzle parameters below the cloud base are retrieved as one of the level 2a Cloudnet products using the algorithm introduced in O'Connor et al. (2005). The retrieval makes use of lidar backscatter and the first three moments of the Doppler radar spectra. The radar data comes from MIRA radar, the same as the one used for our retrieval. The lidar backscatter information is obtained from an independent observation with a different intrument, i.e. a CHM15X ceilometer.

The drizzle property that we can directly compare is the liquid water path of the drizzle below the cloud base, displayed

in Fig. 12. The graph demonstrates a strong correlation between the two sets of LWP values, spanning a few orders of magnitude. The points align well with the one-to-one line and the correlation coefficient is found to be 0.99. The mean values of $\log_{10}(\mathrm{LWP})$ are -2.15 (ours) and -1.87 (O'Connor et al. (2005) method) with an RMSD of 0.29.



## 6  Summary

We developed a method to simultaneously profile cloud and drizzle properties by exploiting the synergy of ground-based radar, lidar and microwave radiometer measurements. This method has the advantage that the (non-)presence of drizzle is inferred from the best-fit to the data, rather than being imposed prior to the retrieval. The lidar forward model simulates the attenuated backscatter not only below the cloud base but also a few hundred meters into the cloud, taking multiple scattering into account. The cloud and drizzle components are distinguished using a droplet size threshold of 13 microns, a choice that is empirically motivated by the results of numerous observational and numerical studies. The combined droplet size distribution of cloud and drizzle follows a bimodal gamma distribution function. To aid the retrieval and to ensure some level of smoothness, the general shape of cloud and drizzle vertical profiles are parametrized based on empirical findings in the literature. The retrieval products include a full set of microphysical parameters (LWC, effective radius, and number density) and the optical extinction coefficient for the cloud and drizzle components.

The cloud retrieval has been tested using synthetic signals generated from LES output. The vertical (along the height) and horizontal (along the time axis) variations in the true microphysical properties are largely reproduced, thereby verifying forward models and the cloud LWC model. On average, the column-averaged effective radius and the column-integrated quantities (LWP and optical depth) are retrieved within 1% of the mean truth, while the number density is retrieved wihin 5%. This indicates that in the absence of radar calibration error, the retrieval method can potentially achieve a high accuracy. From this LES exercise, it appears that when the radar calibration is inaccurate, assuming a single (incorrect) value of the shape parameter $\nu$ in the retrieval can introduce a significant bias. When $\nu$ is set as a free parameter in the fitting, temporal noise increases but the mean bias in the retrieved cloud properties decreases. A radar systematic error of 3 dBZ leads to a mean bias of $\sim 15\%$ in LWP, $r_e$, and $N$ and this value becomes higher when $\nu$ is fixed to the true value.

The retrieval algorithm is applied to a dataset collected during the ACCEPT campaign in Cabauw, The Netherlands. Single layer liquid water clouds with a varying amount of virga are selected. The clouds are between 200-400 m thick with a LWP varying mostly between 10 and 100 g/m$^2$. The cloud and drizzle LWP are positively correlated, with drizzle LWP about two orders of magnitude smaller. The effective radius of the cloud droplets is found to be less than 5 microns on average, far lower than the threshold value of 13 microns. The mean drizzle effective radius at the cloud base where it is expected to be the largest amounts to $\sim 22$ microns, but can be as high as $\sim 60$ microns in intense drizzle periods. The cloud number concentration averages to about 549 cm$^{-3}$, around four orders of magnitude larger than the drizzle number density. Such a ratio of cloud to drizzle number density is comparable to what is measured for marine-stratocumulus clouds in the MASE-II experiment (Lu et al., 2009).

Different elements of the ACCEPT retrieval products are assessed through comparisons with the results of three independent retrieval methods. The first method relies on the lidar depolarization signal to retrieve cloud properties at 100 m above the cloud base. The second one uses radar Doppler spectra to quantify drizzle reflectivity within the cloud boundaries. Lastly, the third technique derives drizzle LWP below the cloud base using information from both radar and lidar. Overall, we find that the




results of our and the three methods are consistent with each other, considering the expected uncertainties and limitations of each technique.

In closing, the application examples of the retrieval algorithm presented here show promising results. Application to datasets with larger size and variety is necessary to establish and improve the validity of the method. Retrieval evaluations using

radiances measured from space or on the ground, and comparison with CCN (cloud condensation nuclei) measurements could be part of the future development. From the computational point of view, there is room for optimization that would lead to a faster implementation of the algorithm and make it more suitable for a large-scale application (currently, it takes roughly 2.5 hours on a dual core, 3 GHz i7 MacBook Pro with 16 GB of RAM to process the 1.2-hour ACCEPT data on Oct 25, 2014 shown in Figures 6 and 7).

**Appendix A:  Drizzle retrieval using radar Doppler spectra**

Kollias et al. (2011a, b) introduce a method to improve drizzle retrieval within stratiform clouds by analysing the higher-order moments of the radar Doppler spectrum. Luke and Kollias (2013) implement this method and show that it works successfully for almost 50% of the spectra close to the cloud top, decreasing to about 15% towards the cloud base. Here we apply the procedure in Luke and Kollias (2013) to the ACCEPT campaign dataset from the chosen time period in Section 4. The aim is

to compare the drizzle reflectivity as derived from this spectral analysis with $Z_{\mathrm{dzl}}$ retrieved using the technique presented in this paper.

The Doppler spectra were acquired using the same cloud radar, so instrument calibration is not an issue. The spectra were recorded every second with a velocity resolution of 0.0825 m/s and a vertical resolution of 30 m. For comparison purposes, we used the time-height coordinate in Fig. 6 such that one composite spectrum corresponds to one time-height pixel. This

means applying a running window of 30 sec at a particular height above the cloud base. The spectra were shifted such that their spectral peaks coincided and were then averaged per velocity bin. The average velocity of the individual peaks was assigned to be the velocity location of the composite spectrum peak. From each composite spectrum, we computed the skewness as an indicator of drizzle presence. Negative skewness suggests that drizzle is the dominant component of the spectrum. Since the spectral decomposition technique is valid only for spectra with positive skewness, we continued to process only the composite

spectra having skewness equal to or larger than 0.1. Here, positive velocities correspond to downward motions (approaching the zenith-pointing radar).

Each spectrum was decomposed by assuming that the portion to the left of the maximum power was entirely due to the cloud –such that it represented the left half of cloud spectrum– and that the drizzle contribution remained to the right side of the spectral peak. The full cloud spectrum was constructed by assuming that the right half followed a Gaussian shape with a

dispersion estimated from the known left portion. Having constructed a complete cloud spectrum, the drizzle spectrum was then produced from the difference between the the cloud and the composite spectra.

With the cloud and drizzle spectra at hand, the reflectivity of each can be simply calculated from the 0th moment. At this stage, we computed the correction factor to compensate for the turbulence as a function of the spectral broadening $\sigma_t$ (Fig. A2





in Luke and Kollias (2013)). For spectra with $\sigma_t$ larger than 0.1 m/s, this correction was applied to the drizzle reflectivity. To preserve the total power, the cloud reflectivity was corrected (reduced) by the same amount.

Fig. A1 presents the results of the procedure above. Panel (a) shows the observed (total) reflectivity computed from the 0th moment of the composite spectra. This is comparable to the reflectivities shown in Fig. 6. Note that the reflecitivities from the spectra are not corrected for gas attenuation, while those from the Cloudnet product are. The cloud base location is marked by the black line as determined from the lidar attenuated backscatter profile. The skewness displayed in (b) is strongly negative around 4 and 5.6 hr coinciding with high reflectivities and a high amount of drizzle below cloud base. Spectral decomposition is not performed for pixels/areas with skewness less than 0.1 and these show up as white gaps in panels (d)-(h).

The mean Doppler velocity shown in (c) demonstrates primarily updraft motion above the cloud base with a few downdraft streaks, most notably around 17.2 hr. The mean velocity below cloud base is strikingly higher than above cloud base. This is indicative of falling drizzle drops that evaporate, cool the air and cause downdraft motions. The reflectivity-weighted mean velocity for drizzle in panel (d) includes air motion, to allow for a consistent comparison between velocities below and above the cloud base. Below the cloud base there is no retrieval so the velocity fields in (c) and (d) are identical. The cloud reflectivity shows similarities to the observed one, suggesting that cloud is the dominant component. In panel (f), the drizzle reflectivities below the cloud base are identical to the observed reflectivities in (a). Visually, there is a continuous transition between the reflectivities above and below the cloud base. In Fig. 11, we show that there is a mean difference of 4 dBZ between the drizzle reflectivities immediately above or below the cloud base (see Section 5.2). Finally, the air motion in panel (g) is determined from the average velocity of the spectral peaks within the 30-sec time interval during the construction of composite spectra, and the width of the composite spectra is presented in panel (h).

*Acknowledgements.* This research is supported by the Netherlands Organisation for Scientific Research (NWO) through the User Support Programme Space Research project ALW-GO/13-20. The ACCEPT campaign was partly funded by the European Union Seventh Framework Programme (FP7/2007-2013) under grant agreement no. 262254 and was carried out in cooperation with the Leibniz Institute for Tropospheric Research (TROPOS), LMU Munich, and METEK GmbH. The authors thank Lukas Pfitzenmaier (TU Delft) for his assistance with the ACCEPT dataset, and Alexander Myagkov and Patric Seifert (TROPOS) for providing the clean radar Doppler spectra. We are also grateful to Ulrich Löhnert for the provision of the original microwave radiative transfer code upon which the microwave forward model used in this work was based, and to Christine Unal for the fruitful scientific discussions.





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



**Table 1.** Lower and upper bounds for the state vector values used in the optimization. The units, when relevant, are given in the square brackets.

| State vector element | Lower limit | Upper limit |
|---|---|---|
| $\nu_{\mathrm{cld}}$ | 2 | 20 |
| $\hat{h}$ (cloud and drizzle) | 0.001 | 35 |
| $W$ (cloud and drizzle) | 0.001 | 1 |
| $N_{ad}$ (cloud; $[m^{-3}]$) | $10^7$ | $5 \times 10^9$ |
| $ft_{\mathrm{cb}}$ | 0 | 1 |
| $ft_{\mathrm{ct}}$ | -1 | 0 |
| $\nu_{\mathrm{dzl}}$ | 1 | 10 |
| $q$ | 0.001 | 0.03 |
| $\alpha_{\mathrm{ic}}\ [\alpha_{\mathrm{cld}}]$ | $10^{-5}$ | $10^{-2}$ |
| $\alpha_{\mathrm{cb}}\ [m^{-1}]$ | $10^{-6}$ | $10^{-4}$ |
| $\alpha_{\mathrm{id}}\ [\alpha_{\mathrm{cb}}]$ | 0.001 | 1 |
| $\alpha_0'\ [m^{-1}]$ | $10^{-10}$ | $10^{-3}$ |
| $p_{\mathrm{cb}}$ | 1 | 3 |



**Table 2.** Values of the LWP, $r_e$, optical depth and the number concentration (as shown in Fig. 5), averaged over the horizontal distance. The RMSD between the truth and the retrieved values is given as the error of the retrievals. The last column is given here for completeness - see discussion in the text.

|  | Truth | Retrieval with $\nu = 5.5$ | Retrieval with optimized $\nu$ |
|---|---|---|---|
| LWP [g/m$^2$] | 171.68 | 171.96 ± 5.31 | 171.64 ± 9.60 |
| Effective radius [$\mu$m] | 20.26 | 20.39 ± 0.23 | 20.44 ± 0.84 |
| Optical depth | 12.68 | 12.62 ± 0.46 | 12.56 ± 0.45 |
| Number concentration [cm$^{-3}$] | 21.26 | 20.30 ± 1.47 | 20.83 ± 5.14 |





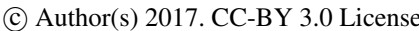

**Figure 1.** The retrieval flow-chart





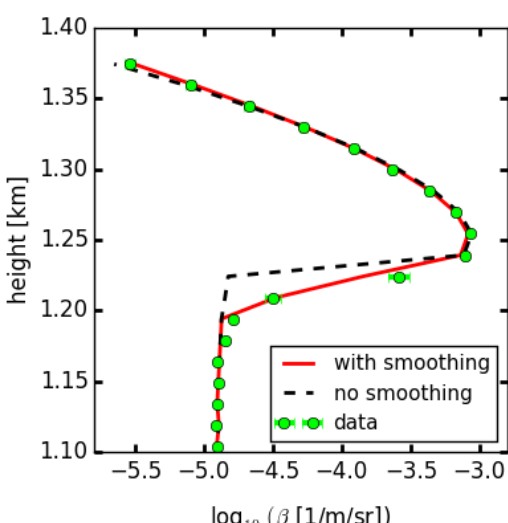

**Figure 2.** Lidar attenuated backscatter profiles as a function of height. The circles outline an example of a measured $\beta$ profile, taken during the ACCEPT campaign on October 26, 2014 at 5.04 hour UTC. The dashed black line shows the forward-modelled $\beta$ that best fits the measurements when no smoothing is applied to LWC. The solid red line represents the forward-modelled $\beta$ when LWC is smoothed.





**Figure 3.** Synthetic signals generated using ECSIM based on LES. The top and middle panels compare radar reflectivity $Z$ and lidar attenuated backscatter $\beta$, respectively, between the synthetic signals that are fed to the retrieval as input (left) and best-fit produced by the retrieval (right). Panels (e) and (f): histograms of the differences between the true and the retrieved signals (truth-retrieval). The spread of the distribution is indicated by the blue vertical lines that mark the interval within which 95% of the total occurences are found. The red curves show the cumulative distributions. Panel (g) shows brightness temperatures $T_B$ averaged over distance at each frequency channel: black circles and line show the synthetic measurements (data mean) while the dashed red line shows the retrieval mean. In (h) we plot the standard deviation of the data mean (black line in (g)) divided by the data mean itself (black) and the RMSD between the retrieval and the data, divided by the data mean (red).






**Figure 4.** True (left panels) and retrieved (right panels) microphysical and optical properties corresponding to the synthetic signal shown in Fig. 3 as a function of vertical and horizontal distance. The first four rows from top to bottom: liquid water content, effective radius, optical extinction coefficient and number concentration. The last row: histograms of the differences between the true and the retrieved cloud properties (truth - retrieved). The spread of the distribution is indicated by the blue vertical lines that mark the interval within which 95% of the total occurences are found. The right y-axes of the four histograms are all identical and correspond to the cumulative distributions shown by the red curves.





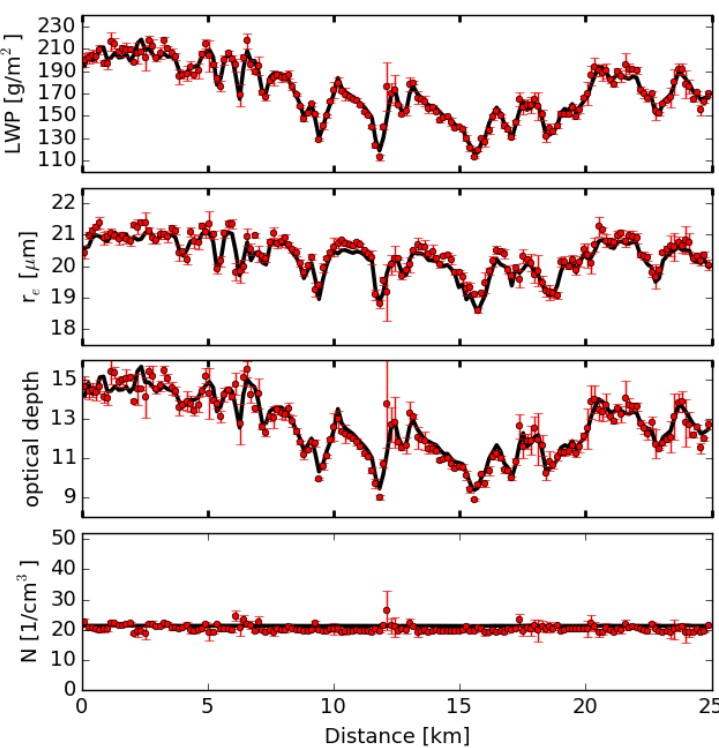

**Figure 5.** Microphysical and optical properties collapsed along the vertical axis: $N$ is column-averaged, LWC and $\alpha$ are integrated into LWP and optical depth, respectively, and $r_e$ are vertically averaged with the corresponding $\alpha$ as the weights. The black line represents the truth and the circles are the retrieved values. The error bars denotes the random errors obtained from the Monte Carlo realizations.



**Figure 6.** Measured and retrieved signals of the selected cases from the ACCEPT campaign. The breaks along the horizontal axes in (a)-(f) mark the change of date from October 25 to 26, 2014. Panels (a)-(d) show the radar reflectivity as observed and as retrieved, along with the decomposition into drizzle and cloud reflectivities. The black line delineates the cloud base determined in the retrieval. Panels (e)-(f) display the observed and the retrieved attenuated lidar backscatter $\beta$. Panels (g) and (h): histograms of the differences between the observed and the retrieved (total) signals (observed - retrieved). The red curves show the cumulative distributions. The spread of the distribution is indicated by the blue vertical lines that mark the interval within which 95% of the total occurences are found. Panel (i) shows the brightness temperatures $T_B$ averaged over distance at each frequency channel: black circles and line show the observations while the dashed red line shows the retrieval. In (j) we plot the standard deviation of the observation mean divided by the mean itself (black) and the RMSD between the retrieval and the observation, divided by the observation mean (red).





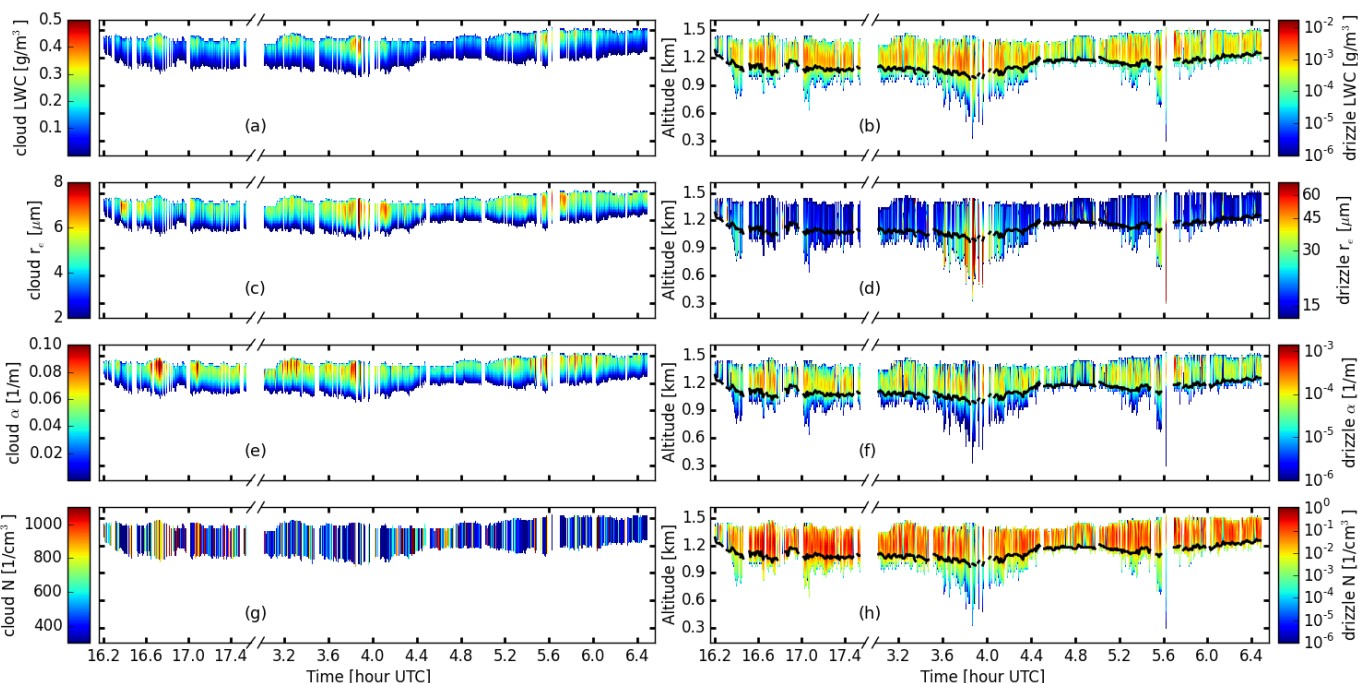

**Figure 7.** Optical and microphysical parameters of the cloud (left) and the drizzle (right) as obtained from the retrieval, as a function of time and height. From top to bottom: liquid water content, effective radius, extinction coefficient and number concentration.



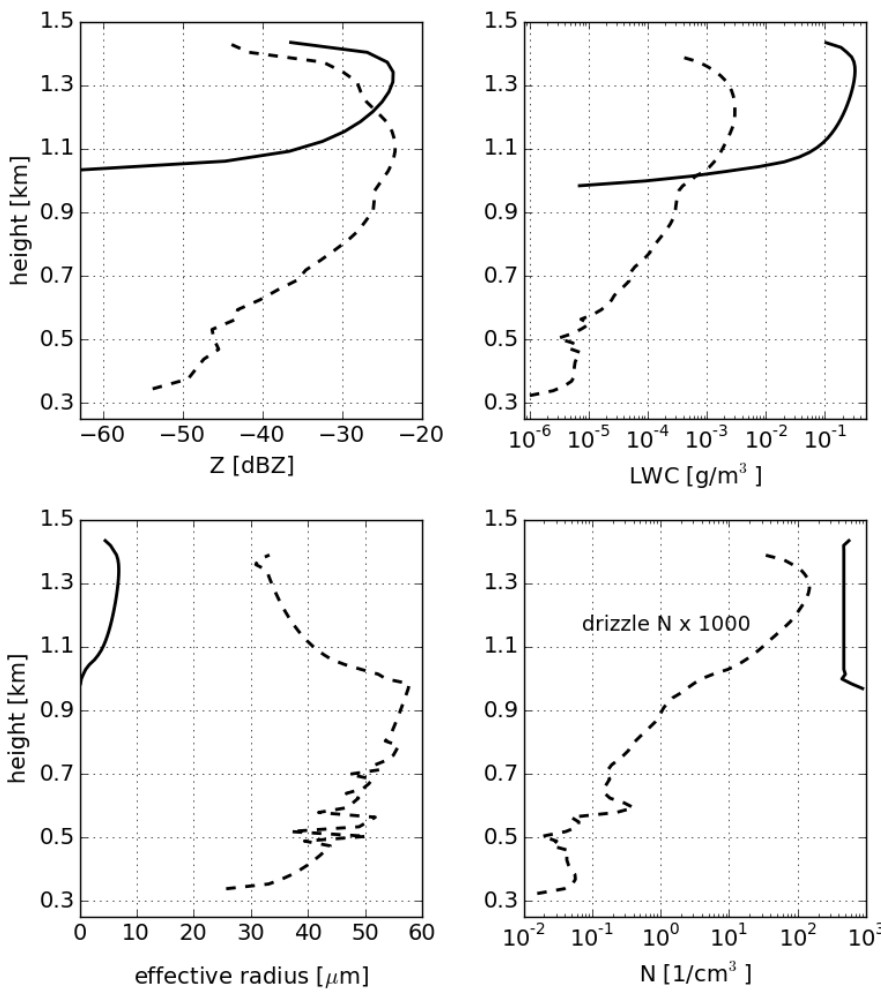

**Figure 8.** Mean vertical profiles of the cloud (solid lines) and drizzle (dashed lines) properties from the observation on October 26, 2014 between 3.8 and 4.0 hr UTC. Clockwise from the top left: radar reflectivity factor in dBZ, liquid water content in g/m$^3$, number density per cm$^3$ and effective radius in microns. Note that the number density of the drizzle has been multiplied by a factor of 1000 for illustration purposes.





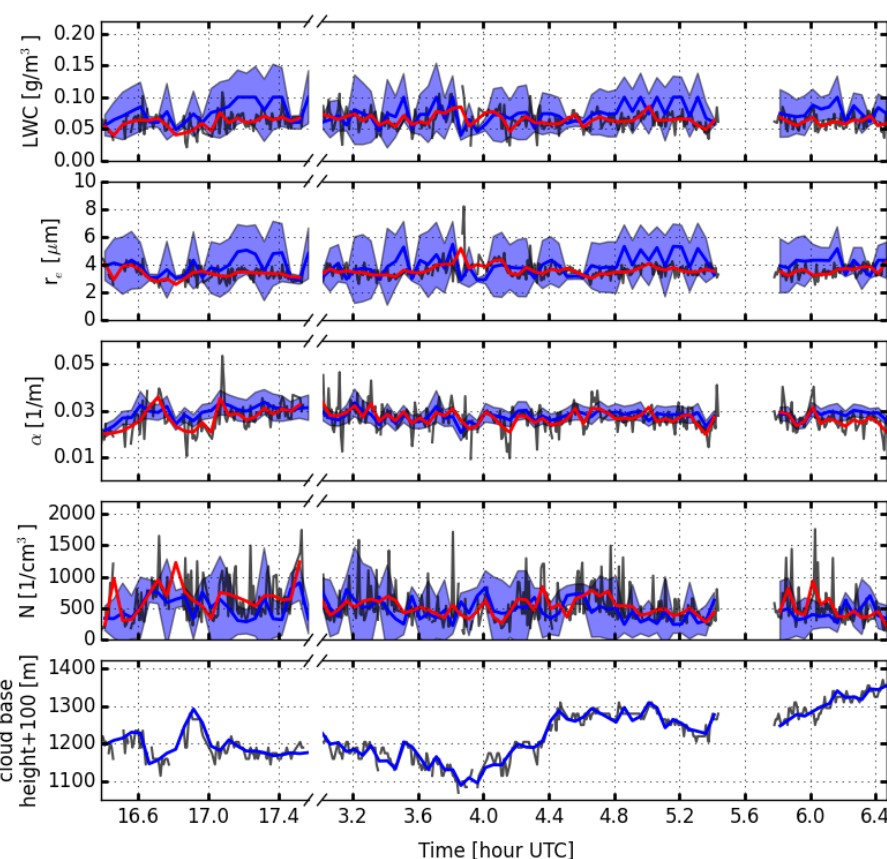

**Figure 9.** Time series of the optical and microphysical properties of the cloud at 100 m above the cloud base. The break along the horizontal axis marks the change of day from Oct 25 to Oct 26. The data around 5.6 hr is deemed unsuitable for retrieval using the depolarization lidar method. The cloud properties and their respective uncertainties derived using the depolarization method are shown by the blue lines and the shaded area. The results of our retrieval are represented by the black lines (temporal resolution: 30 sec) and red lines (time average, to match the time stamps and the 180-sec resolution of the depolarization results). Top to bottom: liquid water content, effective radius, extinction coefficient, number concentration and the altitude at which the cloud properties are evaluated.



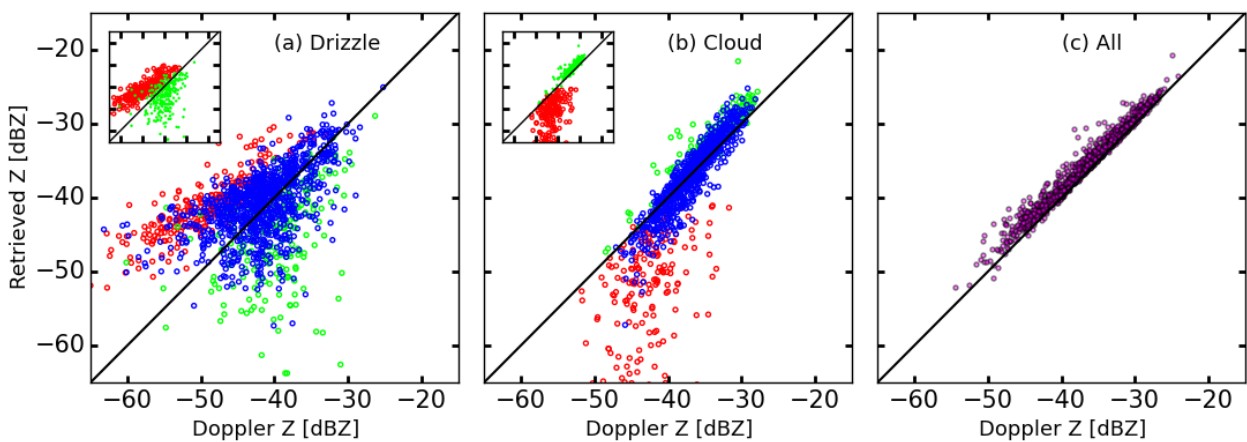

**Figure 10.** Radar reflectivity from our retrieval ('retrieved Z') plotted against those derived from the Doppler spectral decomposition ('doppler Z'). Drizzle, cloud and total reflectivities are shown separately in (a), (b), (c), respectively. The points are color-coded according to their location in the cloud. Red points are from the cloud base area, blue points from the middle of the cloud and green points are from the cloud top region (see the text for details). The insets in (a) and (b) show the same plot with the blue points excluded. The diagonal line is the one-to-one line.





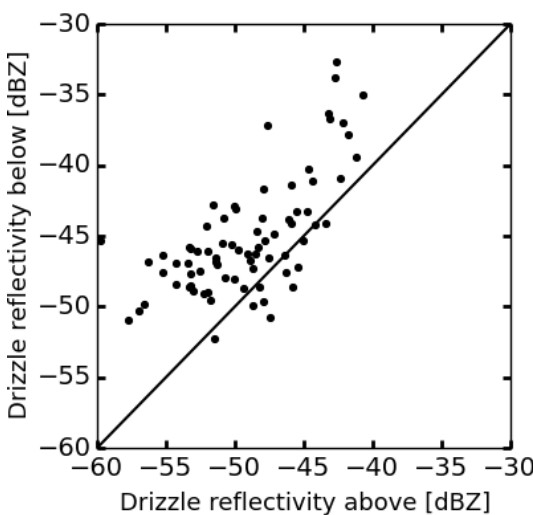

**Figure 11.** Comparison between the drizzle reflectivities at one range gate above and one range gate below the cloud base. The reflectivies above the cloud base are obtained from the Doppler spectral decomposition. The ones below the cloud base are simply the observed reflectivity. The diagonal line marks the one-to-one correspondence.





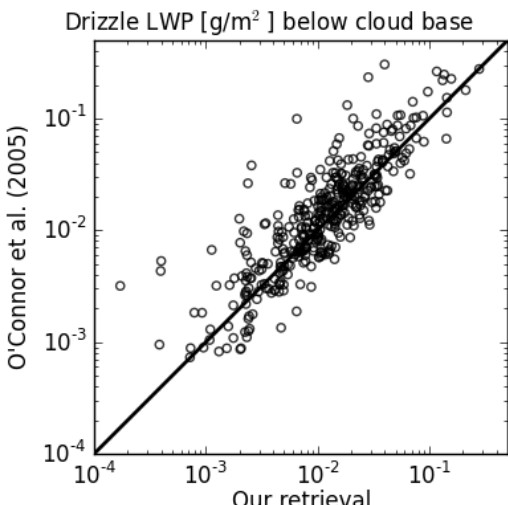

**Figure 12.** Comparison with the drizzle liquid water path below the cloud base as retrieved using the technique of O'Connor et al. (2005), available as one of the Cloudnet products. The diagonal line marks the one-to-one correspondence.



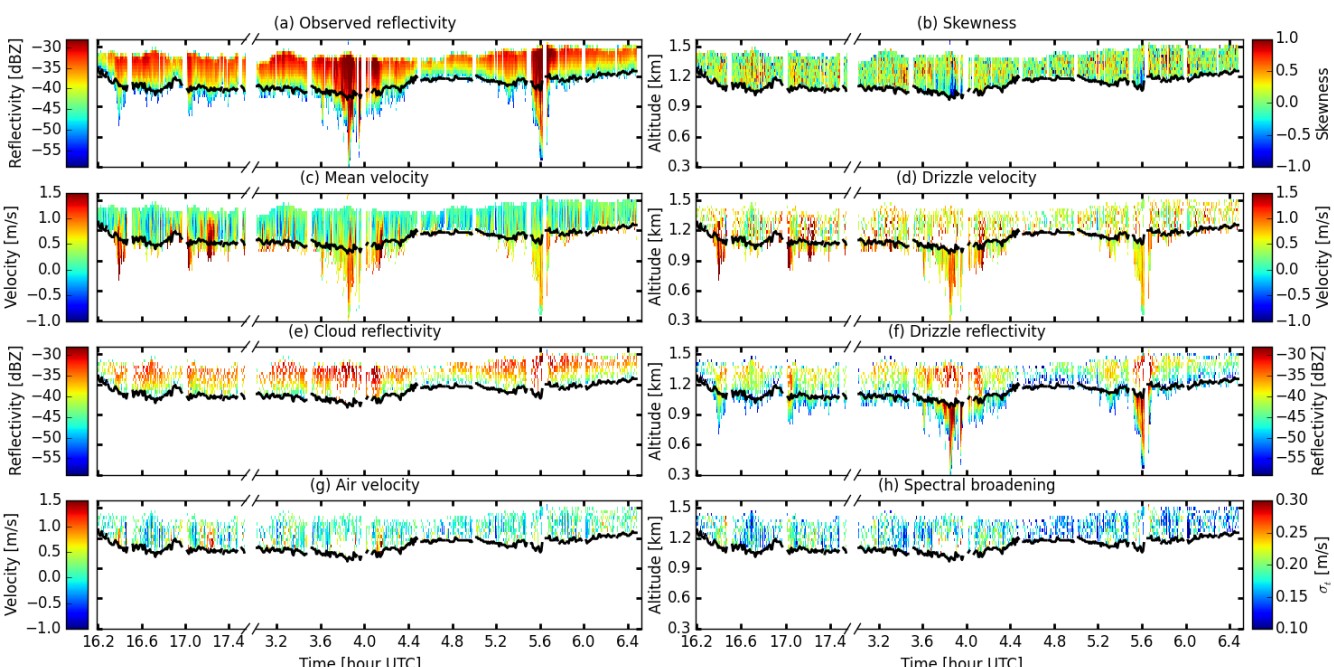

**Figure A1.** Results of the retrieval using Doppler spectra data on Oct 25 and 26, 2014: (a) zeroth moment of the Doppler composite spectra, (b) skewness of the composite spectra above the cloud base, (c)-(d) first moments of the composite spectra and the drizzle spectra, (e)-(f) 0th moments of the cloud and the drizzle spectra, respectively, as obtained from the spectral decomposition, (g) air velocity and (h) spectral broadening due to turbulence. The breaks along the time axis mark the change of day. The black line delineates the cloud base.