# Peer review of "Simultaneous and synergistic profiling of cloud and drizzle properties using ground-based observations"

_Atmospheric Measurement Techniques, 2016_

## Referee Comment (RC1) · Anonymous Referee #1 · 6 Feb 2017

The authors present a novel retrieval method with the goal of solving a long-standing problem of separating the cloud and drizzle property profiles within boundary layer clouds. The retrieval distinguishes itself from pre-existing methods through its unique combination of instruments and physically-based constraints (particularly the vertical structure of drizzle). The ideas presented are therefore clearly suitable for publication in AMT. However, after reading through the manuscript, I have two concerns that could have substantial influence on the results and conclusions.

1) The calibration of the MIRA-35 radar

The observed radar reflectivity reported in Figure 6 is significantly less than I would expect for drizzling stratocumulus. As the authors outline on P2 L25, the consensus

in the literature of typical radar reflectivity for the onset of drizzle is between -20 to -15 dBZ. Yet, on P17 L17, the authors report observed radar reflectivity is typically no higher than -28 dBZ. Given that drizzle is clearly present in Figure 6, and almost reaches the ground at around 4 UTC, I anticipate there is a calibration error of at least 10 dB. Extrapolating the 3 dB error investigated on P16 L4, the retrieved LWC and effective radius are likely to be significantly underestimated. It is therefore difficult to trust the conclusions of the evaluations against other retrieval methods in Section 5. Are there any independent observations of radar reflectivity at Cabauw that could be used to validate the MIRA-35 calibration?

**2) Test using synthetic data**

It is a shame that the LES used to verify the retrieval does not contain drizzle (P14 L22). As the novel aspect of the algorithm is to separate cloud and drizzle signals, the test does nothing but serve as a sanity check to the forward models (in the authors' words on P14 L25) and therefore adds little to the paper. Perhaps testing with idealized profiles of cloud and drizzle would be more informative, or the addition of a synthetic drizzle profile to the LES data? The description of the retrieval technique (Section 2) is somewhat hard to follow, so illustrated examples of the different retrieval scenarios using idealized profiles might be helpful.

Minor comments and style comments:

P1 L18 aerial -> areal

P2 L3 settle -> form

P2 L21 It is not clear whether 'This retrieval' refers to Fielding et al., or the method presented

P3 L12 'respectively' is not needed

P3 L21 define 'Heavy precipitation events'.

P8 L28 minute -> small

P9 L12 If the vertical structure of drizzle within cloud is constrained by Eq. 13, why does the retrieved cloud extinction need to be fixed at 150m? Would it be clearer to include k1 in the state vector (in place of the cloud extinction) and say that any lidar backscatter further than 150m above cloud base is not forward modeled?

P21 L8 (and in other places) when comparing differences in radar reflectivity the unit is dB (relative) rather than dBZ (absolute).

СЗ

---

## Referee Comment (RC2) · Anonymous Referee #2 · 30 Apr 2017

This manuscript describes a retrieval method that provides simultaneous cloud and drizzle retrieval from combined ground-based radar/lidar and microwave radiometer measurements. Since the retrieval presented here is important for understanding cloud and precipitation evolutions, I strongly support the work and admire that the authors made efforts on such a difficult problem. However, I feel that the writing and organization have made the manuscript quite difficult to follow. In particular, I was hoping to provide more specific suggestions how to reorganise the method part, but the unclear and convoluted sentences are just too confusing. I am afraid that the authors really need to rewrite a lot to make sure that the paragraphs have clear and better connections; the sentences/wording are precise and accurate, and more importantly,

many parts sound like "tuning" exercise, when the authors could have provided more convening reasoning.

1) Methodology

There are A LOT of assumptions, retrieval variables and tuning in the proposed retrieval method. Observations are supposed to provide "evidence" of cloud and drizzle profiles to allow us to explore new features, or to test if current assumptions and parameterizations are appropriate in models. If possible, we should let observations speak themselves, rather than forcing all kind of assumptions in the retrieval process. Although the authors mention that these assumptions are based on some other independent observations, it would be good to keep in mind that these assumptions are based on very limited observations, and may not work everywhere. As the authors may have already realised, "adjustment" of these assumptions is needed when this algorithm is applied to different cloud regimes. It would be good to know where the assumption fails, and how this failure affects the overall retrieval. Any limitation does not undermine the value of the proposed work/method.

These assumptions also intrinsically introduce many variables to be retrieved. We need to keep in mind that we only have lidar backscatter, radar reflectivity, and microwave temperature measurements. These are limited observations after all, so we should ask ourselves if these observations really contain sufficient information content to retrieve all the variables proposed in the manuscript. The answer is clearly, a No, and that's why the authors use "tuning" so often in the manuscript. In the end, it will be quite hard to track/ensure that there is no compensating error in the retrieval process. Could the authors comment on this and perhaps have a way to prevent the compensating errors?

The algorithm looks a bit unnecessarily complicated to me. For example, I don't quite understand why is needed to go through all the trouble to "tune" cloud base height. As shown in Figure 2, the authors apply an ad-hoc smoothing in order to get reasonable cloud base height that matches with lidar measurements. Why not using the "observed"

[Figure]

cloud base height instead, and then discuss/understand how sensitive the retrieval will be to the accuracy of the observed cloud base height?

There are also a lot of ad-hoc smoothing bits and thresholds in the proposed method. Rigorous scientific justifications about their choices are needed. For example, why choosing only 1 or 2 radar range gates to classify non-drizzling case. When should we use 1, and when to use 2 gates? Does this really perform better than threshold-based approaches, or they actually agree to a large extent?

Page 5: The justification of constant cloud droplet number concentration (N) with height is a bit misleading, and I feel that the authors are stretching this a bit too far. There is really no sufficient information to infer the vertical profile of N from radar/lidar/microwave measurements. Yes, some could probably use a stronger priori, but the result will not be mainly determined by observations. Saying that a constant N is "adopting the homogeneous mixing case" is just not quite right.

2) Evaluation:

Synthetic data set:

The key point of the manuscript is about cloud/drizzle properties. I am surprised that the authors chose a non-drizzling case in the synthetic data test. Without the presence of drizzle particles, I am less convinced about the performance of the proposed method. I think demonstrating a drizzling case is necessary.

For the current case, I am not sure I understand the results. In section 2, the authors keep emphasizing that they apply a droplet size threshold to separate the cloud and drizzle regime. As a result, they use 13 microns as the separation threshold, meaning that "at any altitude, cloud effective radius has to be smaller than 13 microns and drizzle effective radius cannot be less than 13 microns (page 11)". If that's the case, how come cloud effective radius in Figure 4 clearly exceeds 13 microns for most altitudes? I also don't understand why the majority of radar reflectivity is greater than –20 dBZ for a

non-drizzling cloud (and interestingly, it is opposite for the case from the ACCEPT campaign; see next). Also, the narrow range of cloud droplet number concentration may not be the best case for testing whether the retrieval method is robust.

The ACCEPT campaign:

Could the authors please modify the range of colour bar of radar reflectivity in Figure 6? It is unclear if radar reflectivity is much higher than -30 dBZ or not. If not, it is surprising to see such low radar reflectivity corresponds to drizzle effective radius up to 60 microns. Also, this time series does not include many precipitating profiles. It would be much better to choose another time period that includes a wide range of precipitating conditions.

Could the authors include any independent datasets for evaluations? For example, compare optical depth as shown in Figure 5?

3) Presentation:

Re-organisation of the section 2: I would suggest starting the section with 2.3.2, and making Figure 1 more understandable and stand-alone. The authors need to refer to Figure 1 in a bit more detail to guide readers to understand the overall structure/flow of the retrieval method. It would be nice to construct Figure 1 into a number of main components, provide an overall flow and linkage of all components in the first paragraph, and then synthesize the details in each component.

Some examples that need better connections and wording:

Page 9, Line 18–21: The sentence was talking about $z_{cb}$ and $z_{peak}$, and then the equation below uses $z_{max}$ and $z_{min}$. After reading the line below equation (17), it is unclear how the equation links to $z_{cb}$ and $z_{peak}$. Also, many methods for determining cloud base height from lidar measurements have been proposed and compared; what has presented in this paragraph is a result of the uncertainty in cloud base height determination. Why not mentioning this to justify what has been done here,

instead of presenting them as "the actual cloud base" and "the model cloud base"? More importantly, what is the implication of the need to find the optimal cloud base height? It is not very good news if retrieval needs such precise determination of cloud base height.

Page 9, Line 29: It is unclear what 'as the maximum number of consecutive range gates around z_cb" means. Do you mean, if z_cb is at range gate # 10, for example, then the maximum number of consecutive range gates is 10? What is the physical justification for this smoothing? Softening certain behaviour to get rid of something does not sound very scientific to me. It would be much more appropriate and convincing if the authors could link this behaviour to some sources of uncertainty/noise for justification.

Page 9, Line 29: Is p_cb the pressure at z_cb? It may be obvious for readers, but all variables should be denoted clearly.

There are also quite a few repetitions. Could the authors please read the manuscript carefully and clean things up?

4) Finally, I feel the manuscript could use a bit more positive attitude/tone - we don't need to play down other people's work to justify our work.

Hope it helps.
* * *

---

## Author Comment (AC1) · 4 Jul 2017

The comment was uploaded in the form of a supplement:
https://www.atmos-meas-tech-discuss.net/amt-2016-402/amt-2016-402-AC1-supplement.zip

---

## Author Comment (AC2) · 4 Jul 2017

The comment was uploaded in the form of a supplement:
https://www.atmos-meas-tech-discuss.net/amt-2016-402/amt-2016-402-AC2-supplement.zip

---

## Author Response (AR1)

We would like to thank the editor for his time and also for his remarks, which we address here. We reproduce the comments individually (printed in italic) and write our response below:

On the side, I must say that the treatment of the radiometer stands out also in other ways. The radar and lidar are discussed and treated carefully, while the radiometer is handled less carefully.

In the retrieval, the radiometer is used to provide constraints only on the column-integrated amount of liquid water, while the radar and lidar data provide measurements at individual range gates. The treatment of the data is a reflection of the amount of information/constraints the data can deliver.

For example, the NWP humidity and temperature data going into the forward model will not be perfect. How does this affect the information provided by the radiometer. The same aspect will also cause correlated "noise", while your corresponding covariance matrix is diagonal. And is the radiometer really so well calibrated that those errors can be assumed to be zero (which your diagonal Sy imply)? It would help to have the thermal noise level included in Fig 3h.

The accuracy of NWP humidity (Q) and temperature (T), as well as the radiative transfer model and the measurement uncertainties affect the recovery of the observed  $T_B$  and consequently the retrieved liquid water path (LWP). Several studies have been devoted to investigate the accuracy of LWP retrieval using radiometer data; it is found to be about 15-30 g/m2 (Marchand et al. 2003, Crewell & Löhnert 2003).

Random calibration errors of the radiometer are included in the diagonal elements of Sy. The diagonal Sy implies that the systematic calibration errors are zero (measurement errors of the different channels are assumed to be uncorrelated). Such assumption is not uncommon for retrievals using HATPRO data (e.g. Loehnert et al. 2004, 2009, Ebell et al. 2013).

Typically one can expect that thermal noise is around 0.1 K or less. In the data shown in Fig. 3h, the brightness temperature uncertainties are larger than this (around 0.5 K or more).

**Is the radiometer measuring the same air volume (and with same horisontal resolution) as the active sensors?**

It is technically and physically not possible to guarantee that all of the sensors measure the same air volume. What we can optimize is the representativeness of the measurement volumes. The field of view of the radiometer used in the ACCEPT campaign is approximately 3.5 degrees (half angle), corresponding to an increase of  $\sim 120$  m for each km. The radar and lidar are located 65 m and 20 m from the radiometer. Considering that our targets are located at  $\sim 1$  km, the air volumes measured by the three instruments certainly overlap. In addition, the measurements are averaged over a 30 second period for a single retrieval, which increases the representativeness. Also, the optical and microphysical properties of stratus clouds tend to be horizontally homogeneous with a correlation length scale up to about a km (Schäfer et al. 2017). All in all, it is unlikely with our set-up that the difference in the measured air volumes becomes a severe limitation.

**The manuscript is already long, but I would in fact prefer to see some test inversions without the radiometer, to get a feeling for how important it is.**

Omitting MWR in the retrieval using synthetic data hardly changes the overall results, which means that the radiometer fulfils a complementary rather than a necessary role. This is, however, not always the case with real data and real instruments that are less idealized. Here, the data quality of the other instruments and also from the radiosonde/NWP has to be considered. For example, inaccurate radar calibration potentially leads to an error in LWC and hence LWP. In this case, the LWP constraint from the radiometer can definitely be used as additional information to the retrieval, which should reduce the retrieval error. However, for this to be useful, one should make sure that the Q and T measurements, and the water vapor & oxygen absorption models can reasonably reproduce the observed brightness temperature at the frequencies where  $H_2O$  and  $O_2$  are dominant. From the 14 frequency channels that we use for example, these frequencies correspond to the 5 lowest and 5 highest frequency channels. For the ACCEPT data, we use the model Q and T from the Regional Atmospheric and Climate Model RACMO and we find that the brightness temperatures at these frequencies are well reproduced. Therefore we include the radiometer data in the retrieval. References:

- Crewell, S., U. Löhnert (2003), Accuracy of cloud liquid water path from ground-based microwave radiometry 2. Sensor accuracy and synergy, Radio Science, 38, 8042.
- Ebell, K., E. Orlandi, A. Hünerbein, U. Löhnert, S. Crewell (2013), Combining ground-based with satellite-based measurements in the atmospheric state retrieval: Assessment of the information content, J. Geophys. Res., 118, 6940.
- Löhnert, U., S. Crewell, C. Simmer (2004), An Integrated Approach toward Retrieving Physically Consistent Profiles of Temperature, Humidity, and Cloud Liquid Water, J. Appl. Meteorol., 43, 1295.
- Löhnert, U., D.D. Turner, S. Crewell (2009), Ground-Based Temperature and Humidity Profiling Using Spectral Infrared and Microwave Observations. Part I: Simulated Retrieval Performance in Clear-Sky Conditions, J. Appl. Meteorol. Clim., 48, 1017.
- Marchand, R., T. Ackermann, E.R. Westwater, S.A. Clough, K. Cady-Pereira, J.C. Liljegren (2003), An assessment of microwave absorption models and retrievals of cloud liquid water using clear-sky data, J. Geophys. Res., 108, 4773.
- Schäfer, M., E. Bierwirth, A. Ehrlich, E. Jäkel, F. Werner, M. Wendisch (2017), Directional, horizontal inhomogeneities of cloud optical thickness fields retrieved from ground-based and airborne spectral imaging, Atmos. Chem. Phys., 17, 2359.

We thank the referee for taking the time to read through the manuscript and for the constructive suggestions and criticisms. Below we reproduce the referee's comments (printed in italic) and address them individually:

**1. The calibration of the MIRA-35 radar**

The observed radar reflectivity reported in Figure 6 is significantly less than I would expect for drizzling stratocumulus. As the authors outline on P2 L25, the consensus in the literature of typical radar reflectivity for the onset of drizzle is between 20 to 15 dBZ. Yet, on P17 L17, the authors report observed radar reflectivity is typically no higher than 28 dBZ. Given that drizzle is clearly present in Figure 6, and almost reaches the ground at around 4 UTC, I anticipate there is a calibration error of at least 10 dB. Extrapolating the 3 dB error investigated on P16 L4, the retrieved LWC and effective radius are likely to be significantly underestimated. It is therefore difficult to trust the conclusions of the evaluations against other retrieval methods in Section 5. Are there any independent observations of radar reflectivity at Cabauw that could be used to validate the MIRA-35 calibration?

We compare the reflectivity values from MIRA-35 radar with those from a colocated 3.3 GHz radar (TARA). TARA was operational during the ACCEPT campaign and was independently calibrated. Due to the difference in radar frequencies, the comparisons are focused on periods with precipitation events which are detected by both radars. Figure 1 below shows the reflectivities at a 1000 m altitude on October 25, 2014 between 13:30 and 15:00 (UTC), just before the period analysed in the manuscript. The left panel of Fig. 1 displays the time series and the right panel shows the scatter plot. TARA measurements were collected with a 45 deg elevation angle, while MIRA was pointing to zenith. At 1000 m, both radars observed different resolution volumes, which explains the large scatter. However, there is no obvious sign of strong miscalibration of MIRA.

Figure 1: Time series and scatter plot of radar reflectivity values measured using MIRA and TARA at 1000 m. TARA was operated with a 45-degree elevation angle, while MIRA was pointing to zenith.

To confirm this, we consider another case from the ACCEPT campaign when both radars pointed to zenith. A two-hour period with light to moderate rain events on October 4 between 19:30 and 21:30 (UTC) is selected. We show the time series and the scatter plot in Figure 2 below. For reflectivities higher than 20 dBZ, the small offset in the scatter plot is due to the different attenuations observed at different radar frequencies. From the two comparison cases, we find no evidence of a significant calibration error for the MIRA-35 radar.

It is perhaps relevant to note that in this work we use an effective radius threshold of 13 microns and the retrieved droplet radius of the drizzle that we detect is mostly between 13 and 25 microns. It is common in observational studies or in-situ measurements to define drizzle as droplets with a higher

---

## Referee Report (RR1)

The authors have addressed my original comments satisfactorily; in particular my concern for the radar calibration has been allayed. However, I have some additional questions and issues that I believe need addressing before the paper is ready to be published in AMT.

1) In my opinion, Figure 1 has been over simplified and no longer fulfills its purpose of aiding the reader to piece the different bits of the retrieval together. I suggest using the flowchart to emphasize the key assumptions and intricacies of the retrieval (e.g., smoothing LWC at cloud base, choosing which assumptions are applied). As the retrieval is quite complex, a good flowchart will go a long way to guiding the reader through the method section.

2) Performance of the retrieval in 'Case I', where drizzle is detected only above cloud base. I am worried that there is not enough information to constrain both drizzle and cloud properties. How sensitive are the forward models to the first guess of drizzle scale factor? For example if the drizzle LWC is doubled or halved in CASE I of Fig 6., and all other settings of the retrieval remain unchanged, does the retrieval give satisfactory results?

3) What causes the 'kink' in retrieved drizzle properties at ~0.5 km in CASE II of Fig. 6?

4) How sensitive are the results of drizzle below cloud base to the lidar backscatter? In CASE II of Fig. 6, the lidar backscatter below cloud base is identical or at least very similar to the clear air backscatter (CASE I Fig. 6). Perhaps a greater LWC of drizzle would allow the synthetic drizzle to be detectable above the molecular backscatter.

5) There is too much discussion of the retrieval in cloud only mode (Sect. 3.1). I suggest removing Figure 3,4 and 5, replacing them with a 'Case 0: cloud only' set of panel plots, similar to those in Fig 6. I think this will make the paper more concise and focused on the problem at hand.

Minor comments:

6) Pg 9, ln 32 The extinction coefficient, effective radius, k1 and k2 are all dependent on each other, so the choice of which to include in the state vector is arbitrary. I suggest removing this sentence.

7) Pg 17 ln 13 is too informal

8) Pg 18 ln 12 Surely a thorough error analysis is exactly within the scope of the paper? I suggest removing this sentence.

---

## Author Response (AR2)

We thank the reviewer for carefully reading the revised manuscript and for providing further suggestions. Please find below our answers and responses to the follow-up questions and suggestions (printed in italic). Corresponding adjustments made on the text in the manuscript are shown in green.

1. *In my opinion, Figure 1 has been over simplified and no longer fulfills its purpose of aiding the reader to piece the different bits of the retrieval together. I suggest using the flowchart to emphasize the key assumptions and intricacies of the retrieval (e.g. smoothing LWC at cloud base, choosing which assumptions are applied). As the retrieval is quite complex, a good flowchart will go a long way to guiding the reader through the method section.*

    Following the comments of the second reviewer in the first round, we simplified the flowchart in an attempt to make it more understandable and to emphasize the overall structure of the retrieval. We have now revised Fig. 1 to show more details as currently requested by the reviewer, while maintaining as few distractions as possible from the overall flow. In the new Fig. 1 in the manuscript, we have added the following: the process of determining cloud boundaries, the procedure to determine drizzle reflectivities, and the branching into case I and case II. For each of these, we put the reference to the relevant section of the text to guide the reader. The text describing the flowchart on page 3 has been adjusted accordingly.

2. *Performance of the retrieval in Case I, where drizzle is detected only above cloud base. I am worried that there is not enough information to constrain both drizzle and cloud properties. How sensitive are the forward models to the first guess of drizzle scale factor? For example if the drizzle LWC is doubled or halved in CASE I of Fig 6., and all other settings of the retrieval remain unchanged, does the retrieval give satisfactory results?*

    We recognize the reviewer's concern about the information content in Case I. We mentioned in the manuscript that our retrieval strategy for Case I is adversely limited due to this (page 9 line 14). We tried to exploit the signal but the little amount of information prompted us to adopt a simple retrieval scheme for the drizzle, i.e. using the cloud LWC parametrization with a scaling factor. Here, our goal is to get an order-of-magnitude estimate of the drizzle quantities, rather than to achieve a high accuracy.

    We have modified the synthetic signals in Case I of Fig. 6 as requested, i.e. by varying the drizzle LWC. Changing drizzle LWC consequently changed its $r_e$, N and $\alpha$ while drizzle Z was kept the same. We also kept the retrieval setting identical to the case I retrieval shown in Fig. 6 (e.g. same random seed that determines the retrieval starting point or 'initial guess'). Fig. A below shows the retrieval results for the two requested LWC profiles, i.e. half and twice the original LWC. Fig. B and C expand on this and display the retrieval results when drizzle LWC is decreased or increased by a factor of 3 and 4, respectively. The layout and legend of those figures are the same as in Fig. 6.

    In Fig. A to C, the lidar backscatter and MWR signals (both truth and retrieval) show no discernible change due to the modifications in $LWC_{dzl}$. The cloud properties are robustly recovered in all cases, within the expected level of uncertainties. Fig. A to C also show that changing the drizzle true properties changes the retrieval results as well, despite identical retrieval starting points, cloud properties and reflectivities. This means that the modifications in drizzle LWC can be tracked by the retrieval procedure through the lidar and MWR signals. Similarly to what is shown in Fig. 6, the drizzle reflectivities in Fig. A-C are better retrieved in the lower half than in the upper half of the cloud. Drizzle LWC, extinction coefficient and number concentrations are all retrieved within one order of magnitude from the truth. The drizzle $r_e$ is admittedly poorly recovered in the 0.25x and $0.33 \times LWC_{dzl}$ runs, where the retrieved $r_e$ spans almost the entire radius range allowed in the retrieval, but it gets visibly better constrained for cases with large LWC. This is because the true extinction coefficient gets larger as the true LWC is increased, leaving stronger traces of drizzle in the lidar signal that help constrain the drizzle retrieval. For this reason, drizzle properties are better retrieved for larger $LWC_{dzl}$ than for smaller $LWC_{dzl}$.

    To give a more complete picture of how the starting point of the state vector could influence the retrieval, we performed three retrievals using different random seeds. These runs are based on the

Case I retrieval shown in Fig. 6. The input and settings of the runs are all identical except for the random seed used by the differential evolution routine. The results are shown in Fig. D, which is essentially a copy of the left part of Fig. 6 (case I) with three additional profiles resulting from the different random seeds/starting points (shown in magenta, green and yellow). They all confirm the results from Fig. A-C. The retrieved clouds match the truth with an accuracy consistent with what is expected from the exercise in section 3.1. The retrieved drizzle LWC, extinction coefficient and number concentration give a good order-of-magnitude estimate of the truth with uncertainties of about 50% for the drizzle $r_e$ retrieval.

We have added a few lines in section 3.2 under 'Case I' in the manuscript to note the limited information content and our expectations for the case I retrieval. If the reviewer would like us to elaborate on it, we can show the discussion and the figures here either in section 3.2 or as a separate appendix in the manuscript.

3. *What causes the kink in retrieved drizzle properties at 0.5 km in CASE II of Fig. 6?*

We mentioned the 'kink' as a 'dip' in the manuscript (page 18 line 9). The kink at 530 m originates from the drizzle Z. Since the drizzle $r_e$ is parametrized, the kink does not appear here. Instead, the smoothness of $r_e$ profile propagates the kink to the drizzle LWC, $\alpha$ and N profiles.

This dip in drizzle $Z$ comes about because the cloud reflectivity between 500 and 600m is not well recovered (the effect is most visible in panels IIe and IIg in Fig. 6). More specifically, the retrieved cloud reflectivities at these heights are slightly larger than the true ('observed') $Z$, resulting in $Z_{\text{excess}}$ = 0 (eq. 19). The smoothing that is applied afterwards to produce $Z_{\text{dzl}}$ (see text below eq. 19) replaces the zeros with non-zero values that are smaller than the neighbouring range gates, creating the kink.

We have added this additional explanation in section 3.2 for completeness.

4. *How sensitive are the results of drizzle below cloud base to the lidar backscatter? In CASE II of Fig. 6, the lidar backscatter below cloud base is identical or at least very similar to the clear air backscatter (CASE I Fig. 6). Perhaps a greater LWC of drizzle would allow the synthetic drizzle to be detectable above the molecular backscatter.*

The drizzle results below the cloud base are very sensitive to the lidar backscatter. The drizzle retrieval scheme below the cloud base relies on inferring the magnitude of drizzle extinction coefficient from the lidar backscatter. Noticeable enhancement of backscatter values (on top of the clear air/molecular backscatter) below the cloud base provides a strong indication that the drizzle extinction coefficient is comparable to or larger than the air extinction coefficient. Conversely, when the lidar backscatter appears identical or very similar to the molecular backscatter, one can be sure that the drizzle extinction is significantly smaller than the molecular extinction (drizzle extinction can never be zero as long as Z is detected below the cloud base). In Case II of Fig. 6, the drizzle extinction coefficient amounts to 5% of the molecular extinction coefficient and contributes very little to the lidar signal.

Larger amounts of drizzle would certainly lead to a more detectable drizzle backscatter. To show this, we modified the synthetic drizzle from Case II, i.e. we increased LWC by approximately a factor of 10 and decreased $r_e$ by almost a factor of 2. This results in drizzle extinction coefficient being almost 20 times larger. One can see the effect on the lidar backscatter signal between 200 m and the cloud base. Fig. E shows the retrieval results for the modified signals. Discrepancies between the retrieval and the truth are largest close to the cloud top where the drizzle is least constrained. The cloud retrievals differ slightly from case II Fig. 6, which leads to the differences in the drizzle retrievals above the cloud base. The drizzle properties below the cloud base are more accurately retrieved now, as expected, due to the stronger drizzle signal in the lidar backscatter.

We have added some text and reorganized the paragraphs under 'Case II' in section 3.2 to address this point.

5. *There is too much discussion of the retrieval in cloud only mode (Sect. 3.1). I suggest removing Figure 3,4 and 5, replacing them with a Case 0: cloud only set of panel plots, similar to those in Fig 6. I think this will make the paper more concise and focused on the problem at hand.*

Section 3.1 is dedicated to test and verify the forward models and the cloud LWC model. Here, we use simulated signals where drizzle is absent in the LES scene and is therefore not considered in the retrieval. We view the discussion in section 3.1 as a necessary element and as a basis for discussing the drizzle retrieval that follows. Making sure that the cloud retrieval is robust is crucial especially because the drizzle retrieval depends on and derives from that.

Fig. 3, 4 and 5 allow the reader to visually assess the 'time' evolution and stability of the retrieval. This is an important aspect considering that the retrieval is performed on a column-by-column basis. Replacing them with a set of panel plots similar to Fig. 6 would defeat that purpose. Furthermore, the comparison of the true and retrieved cloud properties for a single profile is already shown in Fig. 6. Since that comparison is representative of the results in section 3.1 as well, it would be redundant to repeat such plots.

We think that categorizing the exercise in section 3.1 as 'Case 0' would be misleading. The actual retrieval scheme as depicted in Fig. 1 does not offer the possibility to ignore/dismiss drizzle prior to the retrieval (as done in Section 3.1), i.e. drizzle is always assumed to be present at the beginning until the optimization process decides otherwise.

For the reasons above, we prefer to keep section 3.1 as it is. However, if the reviewer still feels strongly about this particular point, we could proceed to place some of the text and the figures in the appendix.

6. *Pg 9, ln 32 The extinction coefficient, effective radius, k1 and k2 are all dependent on each other, so the choice of which to include in the state vector is arbitrary. I suggest removing this sentence.*

The sentence has been removed.

7. *Pg 17 ln 13 is too informal*

The sentence has been improved to include a more quantitative evaluation.

8. *Pg 18 ln 12 Surely a thorough error analysis is exactly within the scope of the paper? I suggest removing this sentence.*

The sentence has been removed.

[Figure]

Figure A: Case I retrievals for the modified drizzle LWC. Panels Ia-Il show the results where the true drizzle LWC is halved while panels IIa-IIl correspond to the run where the true drizzle LWC is doubled. The red dash-dotted lines mark the cloud base height $z_{\text{cb,opt}}$. Filled circles: synthetic signals, squares: the cloud truth, diamonds: drizzle truth; red solid lines: retrieved signals, blue dashed lines: retrieved cloud properties and blue dotted lines: retrieved drizzle properties.

[Figure]

Figure B: Case I retrievals for the modified drizzle LWC. Panels Ia-Il show the results where the true drizzle LWC is multiplied by one third while panels IIa-Ill correspond to the run where the true drizzle LWC is tripled. Legend: see the caption of Figure A.

[Figure]

Figure C: Case I retrievals for the modified drizzle LWC. Panels Ia-Il show the results where the true drizzle LWC is multiplied by one fourth while panels IIa-IIl correspond to the run where the true drizzle LWC is increased by a factor of four. Legend: see the caption of Figure A.

[Figure]

Figure D: Case I retrievals with different random seeds. Panels Ia-Il are a reproduction of the same panels in Fig. 6 with the additional green, magenta and yellow lines displaying the retrieval results using different random seeds. Legend: see the caption of Figure A.

[Figure]

Figure E: Case II retrievals. Panels Ia-Il are a copy of the same panels in Fig. 6, repeated here for comparison. Panels IIa-IIl show the retrieval results where the drizzle extinction coefficient is increased by a factor of ∼ 20. Legend: see the caption of Figure A.

[revised manuscript text omitted]
_{peak}$. $z_{cb,opt}$ is located somewhere between $z_{cb}$ and $z_{peak}$, such that:

$$z_{cb,opt} = z_{cb} + ft^*_{cb}(z_{peak} - z_{cb}), \tag{16}$$

where $ft_{cb}$ is constrained to the range (0 ,1). In practice, when the broadening effect is clearly larger than the lidar range resolution then one can set the possible range of $ft^*_{cb}$ to [0,1] and write:

$$z_{cb,opt} = z_{min} + ft^*_{cb}(z_{max} - z_{min}). \tag{17}$$

$z_{min}$ and $z_{max}$ are $z_{cb} + \Delta z_l$ and $z_{peak} - \Delta z_l$, respectively, with $\Delta z_l$ denoting the lidar range resolution.

Once the cloud LWC profile is set up, the smoothing is applied to the region around $z_{cb,opt}$ via the centered moving average scheme. The width of the smoothing window is $2n+1$ where $n$ is the number of lidar range gates between $z_{cb}$ and $z_{cb,opt}$. The LWC values within the smoothing window are weighted as $\exp(-p^*_{cb}d)$. $p^*_{cb}$ acts as a coefficient of the exponential weight and is part of the state vector, whereas $d$ is the distance in the unit of range gates, such that $d = 0$ for the central value, $d = 1$ for the values next to it, and so on. The smoothing is performed only up to $n+1$ gates above $z_{cb,opt}$. Above this height, the impact of the smoothing is insignificant: as $LWC_{cld}$ increases up to the peak value in an approximately linear fashion, the effect of the smoothing quickly diminishes with height.

After the smoothed LWC profile is available, $N^*_{ad}$ and $\nu^*_{cld}$ (both are assumed to be constant with height) given in the state vector can be used to derive profiles of the other cloud properties, i.e. the cloud droplet number concentration (eq. 10 or 11), the extinction coefficient (eq. 3) and the effective radius (eq. 2). In total, seven variables in the state vector are used to construct the profiles of cloud properties: $\nu^*_{cld}, W^*_{cld}, \hat{h}^*_{cld}, N^*_{ad}, ft^*_{cb}, ft^*_{ct}, p^*_{cb}$.

**2.2.2 Drizzle profile**

From the properties derived in the previous section, the radar reflectivity of the cloud component $Z_{cld}$ can be computed (eq. 5). The difference between $Z_{cld}$ and the observed reflectivity $Z_{obs}$ is recorded as $Z_{excess}$, such that:

$$
\begin{aligned}
Z_{excess} &= Z_{obs} - Z_{cld} & \text{for } Z_{obs} > Z_{cld} && (18) \\

[revised manuscript text omitted]